# Intervene When It Doubts: Conjunction-Guided Interactive Reasoning

**Qianyue Wang** [1 2 *]  **Jinwu Hu** [1 2 *]  **Yaofo Chen** [1]  **Yufeng Wang** [1 3]  **Bailin Chen** [1]  **Huanxiang Lin** [1]
**Yu Rong** [4]  **Yuanqing Li** [1 2]  **Zhiquan Wen** [1 †]  **Mingkui Tan** [1 5 †]

## Abstract

Large Reasoning Models (LRMs) excel at complex reasoning but suffer from inefficient reasoning, notably overthinking and overshoot. These issues stem from excessive or misdirected reasoning triggered by the model's "doubt", which manifested as self-validation and exploratory extension, thereby increasing computational cost and degrading performance. Existing efficient-reasoning methods regulate reasoning via internal signals or static schedules, but they are not tailored to the doubt-related characteristics of LRMs. To address this, we propose a **C**onjunction-**G**uided **I**ntervention (**CGI**) reasoning framework that intervenes when the model shows signs of doubt. Our key insight is that overthinking and overshoot in LRMs arise from conjunction-triggered extensions where LRMs signal "doubt" through transitional conjunctions, and then extend redundant self-validation or exploration without timely state-based correction. Building on this insight, CGI pauses reasoning at conjunction-based markers of doubt and injects external state-based feedback, adaptively extending or terminating reasoning to reduce redundancy while preserving accuracy. The feedback is generated through criterion-based evaluation of rationality and completeness and comes from either human or LLM proxies. We train the target model with Group Relative Policy Optimization (GRPO) to adapt to the interactive reasoning mode. Experiments show that our framework achieves a superior balance between accuracy and reasoning length.

---

*Equal contribution  [1]South China University of Technology, China [2]Pazhou Laboratory, China [3]Peng Cheng Laboratory, China [4]DAMO Academy, Alibaba Group, China. [5]Key Laboratory of Big Data and Intelligent Robot, Ministry of Education, China. Correspondence to: Mingkui Tan <mingkuitan@scut.edu.cn>, Zhiquan Wen <sewenzhiquan@gmail.com>.

*Proceedings of the 43rd International Conference on Machine Learning*, Seoul, South Korea. PMLR 306, 2026. Copyright 2026 by the author(s).

## 1. Introduction

Large Language Models (LLMs) (OpenAI, 2023) have become the backbone of natural language processing. While effective across many tasks, they struggle with complex reasoning that requires multi-step deduction and reliable intermediate conclusions (Yang et al., 2024b). To address this limitation, Large Reasoning Models (LRMs) (Huang & Chang, 2023) such as OpenAI o1 (Xu et al., 2025) and DeepSeek-R1 (Shao et al., 2024) have emerged. LRMs separate a reasoning stage from the final *answering* stage, allowing the model to spend additional computation on step-by-step reasoning before a final answer (Zhang et al., 2025a). This design has been shown to improve performance on reasoning-intensive tasks, including challenging mathematics (Art of Problem Solving, 2024), formal logic (Yao et al., 2024), and complex coding (Jain et al., 2025).

However, LRMs reasoning is inefficient, leading to token waste and degraded performance (Sui et al., 2025). Two failure modes are prevalent in inefficient reasoning: *(1) Overthinking:* the model continues with redundant self-verification or exploration despite a correct final answer, inflating latency and computation (Chen et al., 2025b; Yang et al., 2026b); *(2) Overshoot:* after reaching a correct answer, further revision or branching introduces contradictions and overwrites the correct answer (He et al., 2025; Cuadron et al., 2025). Both failures stem from a shared limitation: LRMs lack test-time control over reasoning depth and direction. Reasoning should be treated as a controllable resource that it should be deepened when beneficial, but stopped or redirected once redundant or harmful (Yang et al., 2025c).

Recently, many methods have been proposed to improve reasoning efficiency in LRMs, which are grouped into two paradigms. **Multi-model collaboration** distributes reasoning across models of different scales, but incurs coordination overhead (Pan et al., 2025; Hu et al., 2025). **Single-model optimization** improves reasoning efficiency within a single model, either via high-cost retraining (Aggarwal & Welleck, 2025; Yang et al., 2025b) or by steering decoding at test time using internal numerical signals (Yang et al., 2026b) or static schedules (Chen et al., 2025a; Yang et al., 2026a). Under both paradigms, LRMs internally determine how far to reason, whether to stop, and whether to revise, which leads to

excessive reasoning extension (Echterhoff et al., 2024) and error propagation under cognitive biases (Yu et al., 2024).

To overcome this limitation, a natural remedy is to introduce external feedback during the reasoning process. While several recent works explored reasoning intervention for controllability (Wu et al., 2025) and reliability (Chen et al., 2025a), their strategies remain poorly aligned with the structure and format of LRM reasoning traces in terms of timing, content, and mechanism for efficient reasoning. Existing interventions are often triggered by fixed schedules (Yang et al., 2026a) or numerical signals (Chen et al., 2025a) without timely intervention, and their feedback content is fixed and purely test-time steering (Yang et al., 2026a), which tends to yield limited and unstable control (Wu et al., 2025). Therefore, an intervention framework tailored to the structure and format of the reasoning process for reasoning efficiency is necessary, enabling timely and accurate intervention in the reasoning process, reducing redundancy (Yin et al., 2024; Chen et al., 2024).

We introduce **Conjunction-Guided Intervention (CGI)**, a reasoning process-aligned interactive intervention framework for improving token-efficiency in the reasoning process of LRMs. Our design builds on two key observations: **Observation 1**: transitional conjunctions act as markers for reasoning extension of self-validation or exploration; **Observation 2**: the improvement of conjunction-triggered reasoning in accuracy is state-dependent. Building on these findings, we design CGI that treats the generation of specific transitional conjunctions as intervention points before self-verification or exploration and integrates external feedback based on reasoning state from LLM proxies or humans to adaptively terminate or extend the reasoning process as needed, thereby aligning with the structure and format of the reasoning process of LRMs and reducing the redundant reasoning that leads to overthinking and overshoot. Finally, to ensure the target reasoning model reliably acts upon provided external feedback, we train the reasoning model with the Group Relative Policy Optimization (GRPO) (Shao et al., 2024) under the proposed interactive reasoning paradigm. Our contributions are as follows:

- We empirically identify that transitional conjunctions act as "doubt" markers for self-validation and exploration in LRMs, and the accuracy impact of such "doubt"-triggered extensions flips with the reasoning state. This explains why overthinking and overshoot emerge when "doubt"-driven extensions prolong redundant reasoning without timely correction.

- We propose a conjunction-guided intervention reasoning framework that intervenes when the model doubts: it introduces state-based external feedback at transitional conjunctions, adaptively adjusting reasoning to

reduce "doubt"-induced reasoning redundancy.

- We develop effective task-agnostic feedback criteria based on rationality and completeness that map reasoning states to four actionable guidance types, supporting feedback from multiple sources, including human evaluators and LLM proxies, while minimizing the evaluation burden and avoiding fixed binary feedback or domain-specific hints.

- We adopt Group Relative Policy Optimization with LoRA to fine-tune LRMs under the proposed interactive paradigm with an LLM proxy to further enhance the stability of the model's response to intervention.

**Conflict of Interest Disclosure.** One author is affiliated with Alibaba Group. This work evaluates and uses Qwen-family models, which are developed by Alibaba. The authors declare this relationship for transparency.

## 2. Related Work

**Efficient Reasoning.** Recent works on efficient reasoning can be grouped into two paradigms: **single-model optimization** and **multi-model collaboration**. **Single-model optimization** improves reasoning efficiency within a single model, either by re-training the model to reshape its reasoning (Muennighoff et al., 2025; Aggarwal & Welleck, 2025; Yang et al., 2025b) or by test-time optimization that steers decoding through internal numerical signals or fixed intervention schedules (Yang et al., 2026b; Chen et al., 2025a; Yang et al., 2026a). For re-training, ReasonFlux (Yang et al., 2025b) constrains reasoning using structured thought templates and learns a policy to allocate compute at inference. For test-time optimization, SEAL (Chen et al., 2025a) extracts a transferable latent intervention vector on hidden states during decoding to suppress redundant patterns and DEER (Yang et al., 2026b) performs early exit intervention by monitoring the model's internal uncertainty signals. **Multi-model collaboration** amortizes computation by delegating cheaper sub-processes to smaller models and reserving the expensive model for verification or correction (Zhang et al., 2025b). For example, SpecReason (Pan et al., 2025) lets a lightweight model speculatively draft intermediate reasoning chunks that the base model selectively verifies and corrects, reducing the reasoning length while maintaining task accuracy. While effective at reducing reasoning length, these paradigms rely on self-referential signals to decide when to stop or revise, limited by self-bias (Wataoka et al., 2024). Unlike the above methods, our CGI introduces state-based feedback at transitional conjunctions to intervene in the reasoning process, adaptively extending or terminating the reasoning to reduce redundancy that can lead to overthinking and overshoot.

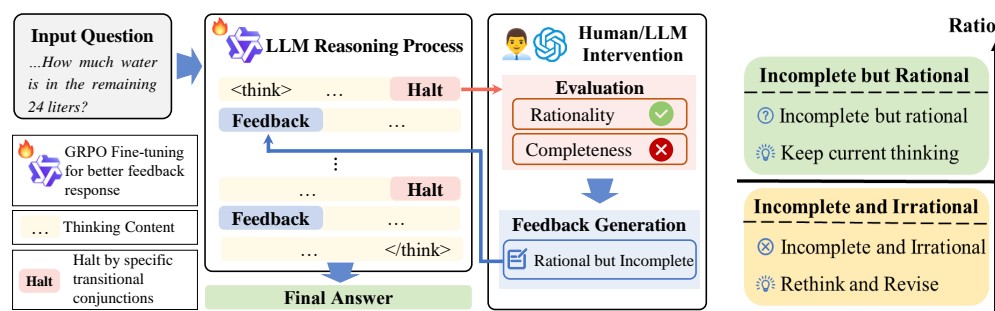

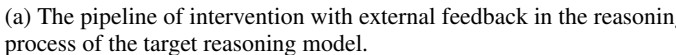

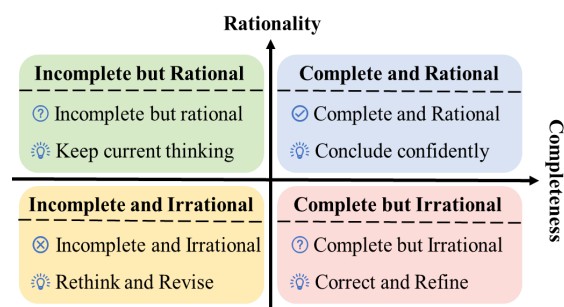

(a) The pipeline of intervention with external feedback in the reasoning process of the target reasoning model.

(b) Four reasoning states by two pre-defined criteria and their corresponding feedback overview.

*Figure 1.* Overview of CGI. Given a question, the LRM in CGI generates a reasoning trace and pauses at predefined transitional conjunctions. An evaluator assesses the current reasoning for rationality and completeness and inserts feedback for subsequent reasoning into LRM's context. The LRM then resumes reasoning with the updated context. This interaction repeats until the model emits `</think>` to self-terminate the reasoning and output the final answer, or is forced to terminate when the interaction limitation is reached.

## 3. Problem Statement

Let $\mathcal{M}_\theta$ be an LLM over vocabulary $\mathcal{V}$ that autoregressively generates tokens by $y_j = \mathcal{M}([x, y_{<j}])$, where $x \in \mathcal{V}^*$ is the input and $y \in \mathcal{V}^*$ is the output. For LRMs, generation is decomposed into a *reasoning stage* to produce a reasoning trace $r = (r_1, \ldots, r_k) \in \mathcal{V}^*$ and an *answering stage* to produce the final output $y$. LRMs often exhibit two inefficient failure modes: **overthinking**, where reasoning is unnecessarily extended and **overshoot**, where continued post-solution reasoning degrades accuracy once the reasoning length $k$ exceeds an instance-dependent optimum $k^*$.

We consider an intervention strategy $\tau \in \{0,1\}^k$, where $\tau_i = 1$ indicates an intervention triggered after generating $r_i$ and observing $r_{\leq i}$. Let $y_\tau(x)$ denote the final output under policy $\tau$. The number of feedback interactions:

$$I(\tau) = \sum_{i=1}^{k} \tau_i. \tag{1}$$

Our goal is to achieve efficient reasoning through the trade-off between accuracy and reasoning length. Since the intervention triggers reasoning extensions, our objective extends to balancing accuracy and the interaction cost:

$$\min_{\tau} \ \mathbb{E}_{(x,y^*)\sim\mathcal{D}}\Big[\mathcal{L}\big(y_\tau(x),\, y^*\big) \,+\, \lambda\sum_{i=1}^{k}\tau_i\Big], \tag{2}$$

where $\mathcal{L}(\cdot,\cdot)$ is the task loss, $\lambda$ controls the cost and accuracy trade-off and $I(\tau)$ measures the interaction cost. where $\mathcal{D}$ is the given dataset and $y^*$ is the ground-truth answer.

## 4. Conjunction-Guided Intervention

In this paper, we propose Conjunction-Guided Intervention (**CGI**), a process-aligned interactive intervention framework that introduces external feedback into the reasoning process to adjust the ongoing reasoning process. CGI addresses three challenges in intervening in the reasoning process: **(1) Where to intervene:** Guided by the semantic role revealed in Observation 1, we treat transitional conjunctions as intervention points in the reasoning process, enabling timely feedback before the model enters the subsequent stage of self-validation or exploratory branching. **(2) What to intervene:** Motivated by Observation 2 that conjunction-triggered extensions have a state-dependent dual effect on performance, we construct criteria for evaluating the reasoning state and map it to state-aware feedback that guides the backbone model to adjust subsequent reasoning, reducing evaluation burden. **(3) How to intervene:** To make feedback distinguishable and executable, we wrap it with tags in the reasoning context. We further incorporate GRPO training via a LoRA adapter to improve the backbone model's stability in responding to feedback under the proposed interactive reasoning paradigm. Tailored to the reasoning in LRMs in precise timing, accurate content, and appropriate mechanism, CGI dynamically adjusts the reasoning process, thereby reducing redundant reasoning and improving efficiency. Figure 1 illustrates the interaction loop of CGI and Algorithm 1 summarizes the pipeline of CGI.

### 4.1. An Information-Theoretic Motivation for State-Based Intervention

We show from an information-theoretic perspective that injecting state-aware feedback reduces the model's uncertainty about the correct reasoning direction, thereby mitigating overthinking and overshoot. Consider an LRM generating a reasoning trace step by step; let $T$ denote the target reasoning direction or final answer, and $C$ denote the current context, with the model's uncertainty given by $H(T \mid C; \theta)$. When the model "doubts" and begins self-validation or exploratory extension, this uncertainty is high, yet the sponta-

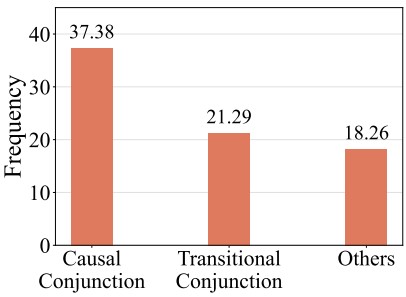

(a) Frequency analysis of semantic tokens among the top 150 tokens on average per reasoning trace.

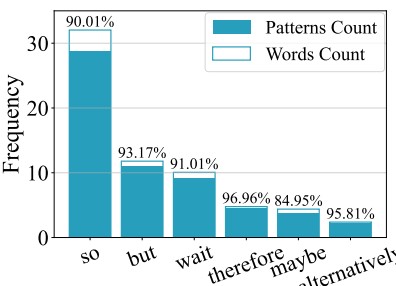

(b) Pattern-word frequency ratio in reasoning traces.

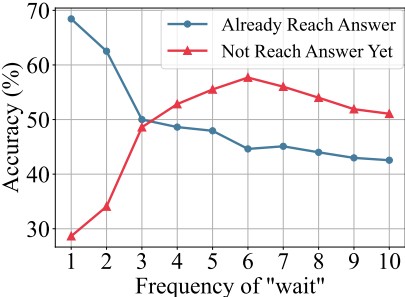

(c) Impact of intermediate thinking correctness on final accuracy.

*Figure 2.* The key observations: (a) Causal and transitional conjunctions rank high among semantic tokens in each reasoning trace. (b) These tokens fit semantic phase boundaries, serving as natural intervention checkpoints. (c) Raising transitional conjunction frequency boosts accuracy before answers, yet impairs performance post solution or with overuse.

neous extension introduces little target-relevant information beyond $C$. Since such extensions provide limited conditional information about $T$, $H(T \mid C;\theta)$ remains largely unchanged, leading to inefficient reasoning (Cover, 1999).

Our framework intervenes at these high-uncertainty moments by inserting external feedback $F$. The mutual information between $T$ and $F$ conditioned on $C$ is non-negative:

$$I(T; F \mid C;\theta) = H(T \mid C;\theta) - H(T \mid C, F;\theta) \geq 0. \quad (3)$$

Therefore,

$$H(T \mid C, F;\theta) \leq H(T \mid C;\theta). \quad (4)$$

The inequality is strict whenever $F$ provides target-relevant information beyond $C$. We generate $F$ by evaluating the reasoning state through *rationality* and *completeness*, two criteria oriented toward reaching the final answer. This design makes $F$ is more likely to be informative about $T$, thereby yielding a strict entropy reduction in practice and reducing redundant reasoning. This entropy-reduction principle underpins CGI's mechanism: by intervening at doubt markers and delivering state-aware feedback, we suppress redundant extensions and preserve efficient reasoning.

### 4.2. Process-Aligned Intervention Points in Reasoning.

Based on the above motivation, the first step of CGI is to identify the intervention points that align with the reasoning process. To identify appropriate intervention points, we first verify that conjunctions are among the most frequent words in reasoning traces, as shown in Figure 2 2a. We further conduct a preliminary study of conjunction-centered language patterns, which leads to the following observation:

**Observation 1: Transitional conjunctions naturally act as markers for reasoning extension of self-validation or exploration.** Prior work uses language patterns to analyze specific reflective behaviors (Liu et al., 2025), and we

---

**Algorithm 1** Pipeline of CGI.

---

**Require:** Reasoning-enhanced LLM $\mathcal{M}_\theta$ , Questions $Q$, Evaluator $\mathcal{E}$, Stop token set $S = \{s_1, \ldots, s_n\}$, Maximum intervention limit $I_{max}$.
1: Initialize set of finished thinking trace $\mathcal{R} \leftarrow \{\}$, set of finished answer part $\mathcal{A} \leftarrow \{\}$
2: **for** each question $q \in Q$ **do**
3:     Initialize generation sequence $gs \leftarrow$ "", intervention count $t \leftarrow 0$
4:     **while** </think> **not in** $gs$ **do**
5:         Generate until reaching any $s_i$ in $S$: $r_t \leftarrow \mathcal{LM}_\theta(q \oplus gs)$
6:         Update $gs$: $gs \leftarrow gs \oplus r_t$
7:         **if** </think> **not in** $gs$ and $t < I_{max}$ **then**
8:             Get external feedback: $f \leftarrow \mathcal{E}(q \oplus gs)$
9:             Construct feedback: $gf \leftarrow$ <reasoning_feedback> $\oplus f \oplus$ </reasoning_feedback>
10:           Update $gs$ with feedback: $gs \leftarrow gs \oplus gf$
11:           $t \leftarrow t + 1$
12:         **else if** </think> not in $gs$ and $t \geq I_{max}$ **then**
13:           Last generate: $r_t \leftarrow \mathcal{LM}_\theta(q \oplus$ </think>$)$
14:           Update gs: $gs \leftarrow gs \oplus r_t$
15:         **end if**
16:     **end while**
17:     Split $gs$ by </think>: $gs_{list} \leftarrow$ Split($gs$, </think>)
18:     Add $gs_{list}[0], gs_{list}[1]$ to $\mathcal{R}, \mathcal{A}$ respectively
19: **end for**
**Ensure:** Thinking trace set $\mathcal{R}$, final answer set $\mathcal{A}$

---

employ conjunction-centered patterns to characterize the general semantic role of conjunctions in structuring reasoning. From Figure 2b, over 90% of conjunctions appear in predefined summarized patterns. Based on the semantics of these patterns, causal conjunctions tend to the reasoning conclusion, and transitional conjunctions tend to trigger self-validation or exploratory extensions to address potential doubt (Sui et al., 2025). Details are in Appendix C.

Motivated by Observation 1, we treat transitional conjunctions as process-aligned intervention points to insert feedback. Let $S$ denote a predefined set of transitional conjunc-

tion tokens used as intervention points. During decoding, once the backbone model emits a token in $S$, the model suspends generation and waits for external feedback before entering redundant self-validation and exploration. Further analysis of different intervention points is in Section 5.4.

## 4.3. Reasoning State-based Feedback

To determine what feedback the LRM needs to adjust its subsequent reasoning, we analyze the effect of conjunction-triggered reasoning extensions and obtain the observation:

**Observation 2: Conjunction-triggered extension has a state-dependent dual effect on task performance.** We segment reasoning traces from DeepSeek-R1-32B at the transitional conjunctions identified above and use Qwen2.5-72B-Instruct to evaluate whether the partial reasoning contains the correct answer and calculate final-answer accuracy at different segment times. As shown in Figure 2c, extending reasoning at transitional conjunctions improves accuracy when a correct solution has not yet been reached, but becomes harmful after the correct answer has been reached or when such transitions are overused. While prior works (Sui et al., 2025; Marjanovic et al., 2026; Choi et al., 2025) report the dual effect of reasoning length on accuracy of reasoning length, we identify the boundary between the beneficial and harmful regimes. This phenomenon generalizes across other challenging benchmarks (details are in Appendix C.3.2).

Inspired by Observation 2, reasoning should be extended when the current trace is insufficient, but should be terminated once the trace is already sufficient. Therefore, external feedback in CGI is generated by task-agnostic criteria based on the current reasoning state: (i) **Rationality**, which checks whether the current reasoning is logically valid and internally consistent (Bai et al., 2022); and (ii) **Completeness**, which checks whether the reasoning has produced a final answer with correct form and intended semantics (Paul et al., 2024). The cross-product of these criteria yields four reasoning states, each mapped to a state-aware, action-specific feedback (This mapping is illustrated in Figure 1b).

Following the criteria, the feedback supports multiple sources, including human and LLM-based proxies, to meet the scenarios constraints (Calderon et al., 2025) and generate feedback with lower evaluation burden and less domain expertise (Lee et al., 2024). Criteria details are in Appendix D.

## 4.4. Enhancing the Responsiveness to Feedback

To improve the backbone model's ability to respond to inserted feedback under the interactive reasoning paradigm, we enhance (i) the feedback injection format and (ii) the model's feedback-following ability. Specifically, we delimit the feedback with explicit tags, `<reasoning_feedback>` and `</reasoning_feedback>` to clearly distinguish gener-

ated reasoning from external feedback within one reasoning context (Shinn et al., 2023).

Instead of applying supervised fine-tuning (SFT) on synthetic reasoning paths, which makes performance sensitive to data quality and incurs potential exposure bias (Ouyang et al., 2022), we adopt Group Relative Policy Optimization (GRPO) (Shao et al., 2024) to improve the model's responsiveness to external feedback under the proposed interactive reasoning paradigm while optimizing the composite accuracy-length objective (Mediratta et al., 2024). For efficient adaptation, we train a LoRA adapter while keeping the backbone parameters frozen.

Following GRPO (Shao et al., 2024) pipline, we sample a group of $G$ candidate rollouts $\{o_i\}_{i=1}^{G}$ from the old policy $\pi_{\theta_{old}}$ through the proposed interactive reasoning paradigm, and an LLM proxy provides feedback at intervention points. Given a stop-trigger token set $S$, the model alternates between reasoning and feedback, producing segments:

$$r_j \sim \mathcal{M}_\theta([x, r_1, z_1, \ldots, r_{j-1}, z_{j-1}]), \quad (5)$$

until backbone model emits `</think>` or reaches the interaction budget and $r_j$ denotes the reasoning segment and $z_j$ denotes the inserted feedback. The generated rollouts are scored by the reward function in Eq. 7 and normalized as:

$$\tilde{\mathcal{R}}_i = \frac{\mathcal{R}_i - \text{mean}(\mathcal{R})}{\text{std}(\mathcal{R})}, \quad (6)$$

We use $\tilde{\mathcal{R}}_i$ as the advantage for all tokens in one corresponding rollout $o_i$ when updating the existing old policy.

**Reward Function.** We design a composite reward to encourage the model to actively follow the injected external feedback under the interactive reasoning paradigm, improving reasoning efficiency (Shao et al., 2024; Dai et al., 2025). For each question–answer pair $(x_i, y_i)$ and a sampled rollout $o_k$ ($k \in \{1, \ldots, G\}$), we define:

$$\mathcal{R}(o_k, y_i) = \mathcal{R}_c(o_k, y_i) + \mathcal{R}_f(o_k, y_i) + \mathcal{R}_l(o_k, y_i). \quad (7)$$

The correctness reward encourages high answer accuracy and provides a signal on task performance. It is defined as:

$$\mathcal{R}_c(o_k, y_i) = \begin{cases} +1, & \text{if the final answer is correct} \\ 0, & \text{otherwise} \end{cases}, \quad (8)$$

The format reward enforces the feedback-aware interaction, preventing the model from ignoring or bypassing feedback:

$$\mathcal{R}_f(o_k, y_i) = \begin{cases} +1, & \text{if format-compliant} \\ 0, & \text{otherwise} \end{cases}, \quad (9)$$

The length reward favors fewer interventions by evaluating

overall reasoning length, including feedback (Team, 2025a):

$$\mathcal{R}_l(o_k, y_i) = \begin{cases} 0.5 - \frac{\text{len}-\text{min\_len}}{\text{max\_len}-\text{min\_len}}, & \text{if correct} \\ \min\left(0, 0.5 - \frac{\text{len}-\text{min\_len}}{\text{max\_len}-\text{min\_len}}\right), & \text{otherwise} \end{cases},$$
(10)

where len is the length of $o_k$, min_len / max_len are the minimum / maximum length of $\{o_1, o_2, ..., o_G\}$.

**Learning Objective.** We optimize the GRPO objective with normalized group advantage $\hat{A}_i$ and KL regularization:

$$\mathcal{J}(\theta) = \mathbb{E}\left[\frac{1}{G}\sum_{i=1}^{G}\left(\min\left(\rho_i\hat{A}_i, \text{clip}(\rho_i, 1-\varepsilon, 1+\varepsilon)\hat{A}_i\right)\right.\right.$$
$$\left.\left. - \beta\,\mathrm{D}_{KL}[\pi_\theta\|\pi_{\text{ref}}]\right)\right],$$
(11)

where $\rho_i = \frac{\pi_\theta(o_i|x)}{\pi_{\theta_{\text{old}}}(o_i|x)}$, $\varepsilon, \beta$ are hyper-parameters and we set $\hat{A}_i = \tilde{R}_i$ as rollout-level advantage, and in implementation this scalar advantage is assigned to all tokens in rollout $o_i$.

# 5. Experiment

**Datasets and Metrics.** We train the target reasoning model on the MATH training dataset (Hendrycks et al., 2021) and evaluate it on benchmarks spanning multiple tasks and difficulty levels, including mathematical reasoning: GSM8K (Cobbe et al., 2021), MATH500 (Lightman et al., 2024), AIME24 (Art of Problem Solving, 2024) and AIME25 (Art of Problem Solving, 2025), multidisciplinary reasoning: GPQA Diamond (Rein et al., 2024), and code generation: LiveCodeBench (Jain et al., 2025). For small-scale datasets, we report AIME24@32 and AIME25@32, where @32 means each problem sampled 32 times and accuracy is computed over all sampled responses (Guo et al., 2025).

We evaluate the performance using: **1) Accuracy (Acc., %)**: The percentage of questions for which the final answer is correct. **2) Average Reasoning Length (Len.)**: The average number of tokens in a complete thinking sequence, encompassing the reasoning content and the external feedback.

**Baselines.** We compare LRM baselines of varying sizes and architectures, including DeepSeek-R1-Distilled-Qwen2.5-7B/14B/32B (Guo et al., 2025), Qwen3-8B/14B/32B and QwQ-32B (Team, 2025b), as well as a prompting baseline that instructs the backbone model to think concisely (Nayab et al., 2024). We compare with efficient-reasoning methods, covering the strategy categories discussed in Related Work. For single-model optimization, we include training approaches L1-Max (Aggarwal & Welleck, 2025) and ReasonFlux-7B (Yang et al., 2025b) and test-time adaptive reasoning strategies, DEER (Yang et al., 2026b) and SEAL (Chen et al., 2025a). For multi-model collaboration,

we include SpecReason (Pan et al., 2025).

**Implementation Details.** The backbone reasoning model in CGI[1] is DeepSeek-R1-Distilled-Qwen2.5-7B, consistent with the target model used by the main training-free baselines. The LLM proxy is built upon Qwen2.5-72B-Instruct following the standard LLM-as-a-Judge evaluation strategy (Li et al., 2024). To guarantee stable termination of the entire interactive reasoning process, we set the maximum number of feedback interactions to 10. For the GRPO training process, we configure the LoRA module with rank set to 8 and the number of generations for each instance is 8, with a max completion length of 8192 tokens and learning rate $1 \times e^{-4}$. The maximum generation length at test time is uniformly fixed to 8192 tokens for all baselines and CGI. More detailed implementation details are provided in Appendix F.

## 5.1. Comparison Experiments

**Superior reasoning efficiency across baselines and reasoning tasks with LLM proxy feedback.** From Table 1, CGI consistently achieves the best accuracy-length trade-off over various baselines and diverse datasets. Compared with the base model DeepSeek-R1-7B, CGI improves accuracy by 33.85% and reduces the average reasoning length by 1802.80 tokens on AIME24. Similar advantages hold across other challenging tasks, such as AIME25, GPQA Diamond and LiveCodeBench. Compared with efficient reasoning baselines, CGI maintains the accuracy advantages while shortening the reasoning length. These improvements arise from the reasoning process-aligned intervention mechanism in CGI. By intervening at these stage-aware checkpoints during the reasoning trace and injecting executable feedback based on the current reasoning state, CGI reduces redundant reasoning that leads to overthinking and overshoot, explicitly balancing the accuracy-length trade-off and yielding efficient reasoning.

**Dynamic control of reasoning length via external feedback across diverse tasks.** From Table 1, CGI adaptively allocates reasoning length by task difficulty while maintaining task accuracy. CGI uses only 394.89 tokens on GSM8K but scales up to 2289.98 tokens on challenging AIME25. In contrast, the backbone model over-generates even on easy tasks and becomes longer on hard ones, reflecting limited test-time control over reasoning depth. This adaptivity stems from effective GRPO training boosting model responsiveness to external feedback, enabling reasoning termination and redirection. These results demonstrate CGI makes reasoning a controllable resource, explicitly balancing accuracy against token cost to improve reasoning efficiency.

**Support multi-source external feedback.** From Table 1

---

[1]The implementation is available in: https://github.com/Qianyue-Wang1/Conjunction-Guided-Intervention.

*Table 1.* Experimental results across baselines and CGI when the external feedback is provided by **LLM proxy**. DeepSeek-R1-7B/14B/32B is the shorthand form of DeepSeek-R1-Distilled-Qwen2.5-7B/14B/32B. **Acc.** indicates task accuracy and **Len.** includes the reasoning content from the target model and the inserted feedback. The same definition applies to all subsequent experiments.

| | GSM8K | | MATH500 | | AIME24 | | AIME25 | | GPQA Diamond | | LiveCodeBench | |
| --- | --- | --- | --- | --- | --- | --- | --- | --- | --- | --- | --- | --- |
| | Acc. ↑ | Len. ↓ | Acc. ↑ | Len. ↓ | Acc. ↑ | Len. ↓ | Acc. ↑ | Len. ↓ | Acc. ↑ | Len. ↓ | Acc. ↑ | Len. ↓ |
| DeepSeek-R1-7B (Guo et al., 2025) | 87.65 | 1110.10 | 84.05 | 1864.33 | 40.00 | 3374.50 | 35.71 | 6371.46 | 22.25 | 5020.86 | 22.29 | 6305.82 |
| DeepSeek-R1-14B (Guo et al., 2025) | 94.05 | 1037.15 | 84.13 | 2003.44 | 53.33 | 3257.70 | 38.89 | 6480.27 | 24.66 | 4411.10 | 28.00 | 6350.63 |
| DeepSeek-R1-32B (Guo et al., 2025) | 95.32 | 1199.42 | 85.98 | 2131.25 | 71.15 | 3134.43 | 44.44 | 8751.76 | 36.30 | 5028.59 | 31.43 | 6387.92 |
| Qwen3-8B (Yang et al., 2025a) | 90.10 | 1379.60 | 86.00 | 2326.29 | 51.97 | 3643.43 | 40.93 | 6254.58 | 22.30 | 5117.45 | 26.29 | 7228.11 |
| Qwen3-14B (Yang et al., 2025a) | 91.96 | 1469.74 | 88.00 | 3014.81 | 62.92 | 7208.37 | 43.44 | 7432.17 | 24.58 | 5694.30 | 32.57 | 6129.63 |
| Qwen3-32B (Yang et al., 2025a) | 94.09 | 1269.77 | 91.00 | 3017.02 | 70.10 | 7009.33 | 49.06 | 7107.67 | 28.23 | 5798.56 | 33.71 | 5979.50 |
| QwQ-32B (Team, 2025b) | 95.52 | 1066.38 | 84.26 | 2540.38 | 66.66 | 4052.80 | 39.68 | 6428.53 | 37.42 | 5585.19 | 32.57 | 6305.63 |
| Prompt Concise (Nayab et al., 2024) | 86.50 | 699.62 | 88.20 | 1259.27 | 52.74 | 2573.43 | 35.56 | 3845.30 | 31.97 | 4120.47 | **34.86** | 5201.16 |
| L1-Max (Aggarwal & Welleck, 2025) | 93.94 | 913.56 | 68.60 | 1615.70 | 30.52 | 6622.96 | 22.19 | 3817.53 | 28.57 | 1731.86 | 24.57 | 3649.40 |
| ReasonFlux-7B (Yang et al., 2025b) | 94.13 | 1296.52 | 90.09 | 3225.43 | 36.36 | 6869.12 | 20.00 | 7427.13 | 26.67 | 5940.47 | 28.81 | 6291.11 |
| SEAL (Chen et al., 2025a) | 89.30 | **297.49** | 84.00 | 2096.87 | 51.35 | 6203.53 | 33.33 | 5919.77 | 57.33 | 3338.80 | 31.30 | 5987.83 |
| DEER (Yang et al., 2026b) | 91.00 | 392.25 | 87.00 | 1280.75 | 54.06 | 4127.76 | 49.79 | 4097.15 | 63.33 | 3136.17 | 30.80 | 3487.17 |
| SpecReason (Pan et al., 2025) | 96.00 | 1075.98 | 92.00 | 2380.78 | 50.00 | 6278.55 | 36.00 | 6431.01 | 50.51 | 5450.72 | 30.00 | 6327.84 |
| CGI (Ours) | **96.21** | 394.89 | **94.00** | 1228.30 | **73.85** | 1571.70 | **71.43** | 2289.98 | 65.22 | 1378.65 | **34.86** | 1560.64 |

*Table 2.* Experimental results across baselines and CGI when **humans** provide feedback in the loop. Considering the human annotation workload, 90 samples are randomly selected from every dataset. For small-scale datasets, we report AIME24@3 and AIME25@3. More details about the human-in-the-loop experiment are in Appendix H and A.

| | GSM8K | | MATH500 | | AIME24 | | AIME25 | | GPQA Diamond | | LiveCodeBench | |
| --- | --- | --- | --- | --- | --- | --- | --- | --- | --- | --- | --- | --- |
| | Acc. ↑ | Len. ↓ | Acc. ↑ | Len. ↓ | Acc.↑ | Len. ↓ | Acc.↑ | Len. ↓ | Acc.↑ | Len. ↓ | Acc. ↑ | Len. ↓ |
| DeepSeek-R1-7B | 88.89 | 1290.73 | 86.67 | 1722.73 | 44.44 | 3903.83 | 38.89 | 6298.03 | 25.56 | 4880.62 | 26.67 | 6274.63 |
| Qwen3-8B | 88.89 | 1731.90 | 88.89 | 3434.68 | 48.15 | 7358.63 | 37.04 | 7402.78 | 59.26 | 6569.07 | 28.00 | 6227.67 |
| ReasonFlux-7B | 94.30 | 1287.95 | 91.11 | 3214.08 | 37.93 | 6717.96 | 22.22 | 7355.86 | 25.93 | 5893.19 | 26.13 | 6331.87 |
| DEER | 91.00 | 392.25 | 86.67 | 1274.44 | 54.00 | 4120.23 | 53.44 | 4108.09 | 57.29 | 3093.31 | 31.90 | 3480.86 |
| SEAL | 88.89 | **300.00** | 83.33 | 2090.69 | 51.11 | 6196.36 | 34.44 | 5889.54 | 56.67 | 3386.66 | 32.20 | 5692.74 |
| SpecReason | **95.56** | 782.94 | 92.22 | 2372.07 | 52.22 | 6205.45 | 35.56 | 6155.71 | 48.89 | 6067.04 | **35.56** | 6337.84 |
| CGI (Ours) | **95.56** | 388.80 | **93.33** | 1254.60 | **61.62** | 1690.54 | **61.11** | 2102.84 | 64.44 | 1216.42 | 34.44 | **1568.50** |

and Table 2, CGI achieves the best overall trade-off between accuracy and reasoning length across different baselines and reasoning tasks whenever the feedback comes from an LLM proxy or humans. Compared with original base model, CGI with LLM proxy improves accuracy by 9.95% and reduces length by 636.03 tokens on MATH500, and CGI with humans improves accuracy by 22.22% and reduces length by 4195.19 tokens on AIME25. This robustness across feedback sources stems from the task-agnostic feedback critics that focus on the reasoning state rather than domain-specific hints, reducing dependence on expertise and evaluation burden, especially for humans (Javernik et al., 2023).

### 5.2. Thinking Behavior under CGI

**GRPO training improves the model's adaptation to interactive reasoning.** From Table 3, CGI achieves over 80% self-stop ratio on most datasets with human feedback and the self-stop ratio ranges from 73.11% to 95.87% with LLM proxy. This self-stop ratio is the fraction of runs where the model autonomously emits </think> to terminate the reasoning and generate the answer, rather than being forced to stop by the interaction budget. These results demonstrate the backbone model in CGI adapts to the proposed interac-

tive reasoning paradigm through GRPO training, making CGI to efficiently balance the accuracy-length trade-off.

**Token and time costs of external feedback from LLM proxy and humans are efficient.** From Table 3, feedback from the LLM proxy accounts for 12% to 25% of total reasoning length, while human feedback is more concise. Both LLM proxy and humans provide feedback within seconds for each complete interactive reasoning process, following the criteria. Intervention frequency adapts to task difficulty. The average numbers of interactions are 0.79 and 0.90 times for LLM proxy and human on GSM8K, while the results are 5.13 and 6.29 on AIME25. The intervention frequency under both settings is within acceptable limits, considering interaction fatigue (Javernik et al., 2023). Details on time and computational costs of methods are in Appendix G.

**LLM proxy feedback and human feedback are suited to different scenarios.** From Table 3, human feedback uses fewer tokens and occupies a smaller share of the reasoning trace, providing targeted guidance which is crucial for the reliability in high-risk or creative cases to improve interpretability (Yao et al., 2023). The LLM proxy produces longer feedback but makes the reasoning process fully automatic, enabling scalable tasks in batch or high-throughput

*Table 3.* Statistics of reasoning generated by CGI. We report the average result per reasoning, and details are in Appendix G.

| Feedback Source | Metric | GSM8K | MATH500 | AIME24 | AIME25 | GPQA Diamond | LiveCodeBench |
|---|---|---|---|---|---|---|---|
| LLM Proxy | average number of interventions ↓ | 0.79 | 2.63 | 3.80 | 5.13 | 3.99 | 4.00 |
| | self-stop ratio (%) ↑ | 95.87 | 93.96 | 76.67 | 73.11 | 79.90 | 74.85 |
| | feedback length (tokens) ↓ | 72.49 | 146.43 | 389.20 | 325.76 | 242.27 | 312.00 |
| | reasoning length (token) ↓ | 322.40 | 1081.87 | 1182.50 | 1964.22 | 1136.38 | 1248.64 |
| | feedback time (s) ↓ | 2.81 | 11.97 | 21.84 | 22.06 | 16.36 | 21.65 |
| | reasoning time (s) ↓ | 8.97 | 24.41 | 37.43 | 37.42 | 20.03 | 18.77 |
| Human-in-the-loop | average number of interventions ↓ | 0.90 | 3.10 | 4.15 | 6.29 | 3.15 | 5.95 |
| | self-stop ratio (%) ↑ | 100.00 | 90.00 | 89.23 | 80.21 | 84.22 | 75.00 |
| | feedback length (tokens) ↓ | 19.00 | 33.10 | 46.77 | 55.85 | 34.63 | 63.65 |
| | reasoning length (token) ↓ | 369.80 | 1221.50 | 1643.77 | 2046.99 | 1181.79 | 1504.85 |
| | feedback time per reasoning (s) ↓ | 2.83 | 13.44 | 27.45 | 60.25 | 18.26 | 34.59 |
| | reasoning time per reasoning (s) ↓ | 8.68 | 17.77 | 28.03 | 54.20 | 17.17 | 25.26 |

*Table 4.* Ablation results of **CGI**. w/o means without and w means with. Details of experiment setting are in Appendix J.

| | GSM8K | | MATH500 | | AIME24 | | GPQA Diamond | | LiveCodeBench | |
|---|---|---|---|---|---|---|---|---|---|---|
| | Acc. ↑ | Len. ↓ | Acc. ↑ | Len. ↓ | Acc. ↑ | Len. ↓ | Acc. ↑ | Len. ↓ | Acc. ↑ | Len. ↓ |
| base model | 87.65 | 1110.10 | 84.05 | 1864.33 | 40.00 | 3374.50 | 22.25 | 5020.86 | 22.29 | 6305.82 |
| w/o feedback tag | 91.13 | 618.22 | 86.40 | 2114.98 | 57.91 | 3592.33 | 46.43 | 2679.55 | 32.57 | 2193.00 |
| w/o fine-grained feedback | 90.36 | 358.56 | 81.57 | 1785.19 | 44.06 | 2177.53 | 43.88 | 1833.67 | 28.65 | **1235.57** |
| w/ base model | 88.09 | **322.95** | 75.23 | **1190.85** | 33.75 | 2151.13 | 37.24 | 2011.89 | 28.00 | 1509.00 |
| w/ SFT model | 90.50 | 399.50 | 86.00 | 1404.15 | 63.33 | 3617.78 | 64.20 | 1515.43 | 28.65 | 2358.55 |
| CGI (Ours) | **96.21** | 394.89 | **94.00** | 1228.30 | **73.85** | **1571.70** | **65.22** | **1378.65** | **34.86** | 1560.64 |

*Table 5.* The impact of different feedback qualities from LLM proxy in CGI. For the proxy model, we use the instruct version of Qwen2.5 in specific parameter scales.

| | AIME24 | | GPQA Diamond | | LiveCodeBench | |
|---|---|---|---|---|---|---|
| | Acc. ↑ | Len. ↓ | Acc.↑ | Len. ↓ | Acc. ↑ | Len. ↓ |
| base model | 40.00 | 3374.50 | 22.25 | 5020.86 | 22.29 | 6305.82 |
| 7B proxy | 59.68 | 1642.33 | 44.95 | 1511.15 | 26.16 | 1549.56 |
| 14B proxy | 61.35 | 1636.40 | 51.53 | 1531.31 | 28.49 | 1521.87 |
| 32B proxy | 63.33 | 1541.48 | 58.67 | 1466.09 | 29.14 | 1544.27 |
| 72B proxy | 73.85 | 1571.70 | 65.22 | 1378.65 | 34.86 | 1560.64 |

*Table 6.* The result of intervention by different strategies in CGI. Setting details are in Appendix K.1. Strategy indicates the intervention points to insert external feedback.

| Strategy | AIME24 | | GPQA Diamond | | LiveCodeBench | |
|---|---|---|---|---|---|---|
| | Acc. ↑ | Len. ↓ | Acc. ↑ | Len. ↓ | Acc. ↑ | Len. ↓ |
| base model | 40.00 | 3374.50 | 22.25 | 5020.86 | 22.29 | 6305.82 |
| every sentence | 51.11 | 2963.37 | 52.67 | 3902.93 | 15.43 | 1911.11 |
| fixed length | 38.33 | 2303.52 | 56.70 | 2360.85 | 25.14 | **1504.77** |
| At "\n\n" | 56.35 | 3943.43 | 62.67 | 1848.23 | 22.14 | 1657.08 |
| conjunctions (CGI) | **73.85** | **1571.70** | **65.22** | **1378.65** | **34.86** | 1560.64 |

settings. Given the flexibility of external feedback sources in CGI, we recommend starting with an LLM proxy and switching to human feedback for difficult or safety-critical tasks. The interface in Appendix I supports this hybrid workflow (Zhou et al., 2026; Pang et al., 2026).

### 5.3. Ablation Study

**Well-delimited and fine-grained feedback is essential for efficient interaction.** Table 4 clearly shows that removing feedback tags (*w/o feedback tag*) or replacing action-oriented feedback with binary static feedback (*w/o fine-grained feedback*) loses the original accuracy-length advantage. On MATH500, removing the tags increases reasoning length from 1228.30 tokens (Ours) to 2114.98 tokens and reduces accuracy from 94.00% to 86.40%; on GPQA Diamond, coarse fixed feedback increases length from 1378.65 to 1833.67 and drops accuracy from 65.22% to 43.88%. These ablation settings make the injected feedback hard to parse and execute, causing the backbone model to spend additional reasoning steps interpreting the feedback and may be disrupted by unnecessary reasoning (Yao et al., 2023).

**Interactive adaptation via GRPO is necessary.** From Table 4, when we ablate GRPO training with two typical alternatives, directly using the unadapted backbone (*w/ base model*) and using an SFT model trained on synthesized traces (*w/ SFT model*), CGI fails to reliably transfer the feedback into reasoning-efficient improvements. On AIME24, *w/ base model* collapses accuracy from 73.85% to 33.75% while *w/ SFT model* also produces much longer traces (3617.78 tokens) with lower accuracy (63.33%). These outcomes point to a shared failure mode that without training under the proposed interactive reasoning paradigm, the model's feedback responsiveness does not generalize across tasks, leading to the risk of overthinking and overshoot.

## 5.4. More Discussions

**Impact of LLM proxy capability on CGI performance.**
From Table 5, CGI consistently improves the accuracy-token trade-off over the base model with LLM proxies of different scales. Notably, even the peer-sized 7B proxy brings clear gains, improving accuracy from 40.00% to 59.68% on AIME24, from 22.25% to 44.95% on GPQA Diamond, and from 22.29% to 26.16% on LiveCodeBench, while reducing reasoning length. As proxy scale increases from 7B to 72B, the overall performance further improves, especially on more challenging benchmarks. These results indicate that the effectiveness of CGI comes from feedback-based intervention in the reasoning process, rather than knowledge distillation. Additional results on potential applications and case studies of CGI are in Appendix L.

**Comparison of different intervention strategies.** From Table 6, intervening at transitional conjunctions (Ours) achieves optimal balanced between accuracy and reasoning length across diverse benchmarks. Across three typical datasets, CGI yields the best steady accuracy gains while achieving competitive short reasoning traces compared with other mainstream intervention strategies. On AIME24, intervening at transitional conjunctions improves accuracy over the "\n\n" strategies from 56.35% to 73.85% with shorter overall reasoning length. This aligns with Observation 1 that transitional conjunctions mark the extension of reasoning for potential redundancy, so feedback is conditioned on a semantically complete reasoning segment. Other strategies either interrupt sub-reasoning prematurely or intervene too late, after redundancy has already emerged, which is not aligned with the semantic extension boundary of LRMs.

## 6. Conclusion

In this paper, we propose CGI, a reasoning process-aligned intervention framework that improves the efficiency of large reasoning models through external feedback. Motivated by the empirical observations of LRM reasoning, CGI pauses reasoning at transitional conjunctions, the reasoning-extension markers for self-validation and exploration and injects state-based feedback from multiple sources to adjust its subsequent reasoning. GRPO training further improves the model's adaptation to the interactive reasoning paradigm. By tailoring the timing, content, and mechanism to the reasoning process, CGI enables dynamically adjustment of reasoning depth. Experiments show that CGI consistently achieves a superior accuracy–length trade-off on multiple reasoning benchmarks. Further discussion of potential limitations and future directions is provided in the Appendix A.

## Acknowledgments

This work was partially supported by the Joint Funds of the National Natural Science Foundation of China under Grant No.U24A20327, the Guangdong S&T Program under Grant No.2026B0101110001, the Guangdong Basic and Applied Basic Research Foundation under Grant No.2026A1515010388, the Postdoctoral Fellowship Program of CPSF under Grant No. GZC20251043.

## Impact Statement

This paper presents work whose goal is to advance the field of Machine Learning. There are many potential societal consequences of our work, none of which we feel must be specifically highlighted here.

## Author Contributions

Qianyue Wang and Jinwu Hu contributed equally to this work. Mingkui Tan and Zhiquan Wen are the corresponding authors. Qianyue Wang conceived the main idea and designed CGI architecture. Qianyue Wang, Bailin Chen, and Huanxiang Lin developed the method and conducted the experiments. Qianyue Wang, Jinwu Hu, Yaofo Chen, Yufeng Wang, Bailin Chen, and Yuanqing Li contributed to implementation, data preparation, evaluation, and result analysis. Qianyue Wang, Jinwu Hu, Zhiquan Wen, and Yu Rong wrote the draft. Mingkui Tan and Zhiquan Wen supervised the project, provided guidance on methodology and experiments, and revised the manuscript.

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

# Supplementary Materials for
## "Intervene When It Doubts: Conjunction-Guided Interactive Reasoning"

## A. Limitation Discussion

While CGI has been validated to improve reasoning efficiency across diverse reasoning tasks, we clarify its operating boundaries and outline practical extensions to support a balanced assessment of its applicability and future development.

**Dependence on feedback quality.** As discussed in Section 5.4, while CGI consistently improves the accuracy-length trade-off over base reasoning models across multiple reasoning benchmarks, the performance of CGI varies with the quality of external feedback from different scales of models. While our criterion-driven feedback design helps maintain goal relevance, inaccurate feedback still triggers inappropriate continuation or revision, leading to ineffective trace prolongation (see the failure case study in Appendix L.2.1).

**Further improving feedback efficiency.** The feedback in CGI is generated by directly conditioning on the current reasoning trace. While criterion-driven feedback design reduces the evaluation burden, a long reasoning trace is necessary for challenging instances (Stiennon et al., 2020), but increases feedback burden and potentially reduces feedback quality. A natural extension is to maintain an incremental summarization of reasoning, so feedback is produced from a shorter representation, further improving interaction efficiency and practicality.

**Potential complexity from the extra proxy.** When feedback is provided by an LLM proxy, CGI introduces an additional model into the inference pipeline. Therefore, although CGI reduces the number of reasoning tokens generated by the target LRM, the extra proxy may increase system complexity, including model serving, memory usage, synchronization overhead, and wall-clock latency. This makes the practical benefit dependent on the deployment setting and the relative cost of the target model and the proxy. A promising direction is to use lightweight or peer-sized proxies, invoke the proxy only at high-uncertainty intervention points, or cache and reuse feedback signals when possible.

**Optional training burden of GRPO adaptation.** Although CGI applies GRPO only to a lightweight additional LoRA module with the backbone model frozen, this adaptation process introduces extra training cost. Table 7 compares the training cost of CGI with representative retraining-based baselines. Notably, GRPO adaptation is not necessarily required for every base model in CGI, since reasoning-enhanced models at different scales have different feedback-following abilities during reasoning process. The interactive intervention framework of CGI itself is model-agnostic.

*Table 7.* Training cost comparison between CGI and baselines.

| Method | Finetuning parameters | Training data volume | Training context | Training algorithm | GPU cost | Iterations |
|--------|----------------------|---------------------|------------------|-------------------|----------|------------|
| L1-Max | 1.5B | 40k | 4k | LCPO | 8×A100 | 300 steps |
| ReasonFlux-7B | 7B | 15k+10k | 8k | HRL+DPO | 8×A100 | 6+6 epochs |
| CGI (Ours) | 1.26M | 7.5k | 8k | GRPO | 4×A800 | 120 steps |

## B. More Related Work

**External Feedback.** Many prior works inject external feedback to additionally constrain LLMs for controllability and reliability, ranging from direct human-preference optimization to tool or critique- based refinement (Stiennon et al., 2020). Such methods differ along (i) *when* feedback is introduced, (ii) *what form* it takes, and (iii) *how* it is integrated into generation (Shinn et al., 2023). From the timing perspective, *pre* strategies expand the prompt with external evidence without conditioning on the model's evolving reasoning state (Lewis et al., 2020). *Post-hoc* strategies produce an initial solution and then iteratively critique and revise it, such as Self-Refine (Madaan et al., 2023) and tool-interactive critiquing in CRITIC (Gou et al., 2024), typically improving output quality at the cost of extra decoding rounds. *In-process* strategies interleave reasoning with immediate textual feedback for LLMs, exemplified by ReAct (Yao et al., 2023), which yields more

interpretable and less hallucinatory trajectories. However, for LRMs that explicitly separate a dedicated reasoning trace (e.g., a `<think>` stage) from the final answer, inference often exhibits overthinking with redundant validation loops and substantial token overhead (Huang et al., 2025); consequently, directly applying generic pre/post-hoc feedback pipelines (prompt expansion or multi-round refinement) can be ill-suited to such reasoning behaviors and may further increase token cost rather than reducing redundant deliberation. However, for LRMs that explicitly separate a dedicated reasoning trace from the final answer, these feedback schemes, which are not fit for the reasoning behaviors, result in additional token cost (Yong et al., 2025). For LRMs, numeric feedback has been used to guide the token generation in the reasoning process of LRM (Setlur et al., 2025) with limited interpretability and flexibility in adjusting *what* to think next. In contrast, CGI specializes external feedback injected within reasoning process of LRMs on timing, format and mechanism. CGI continuously observes the ongoing reasoning process and provides state-aware textual feedback about the current reasoning state, thereby dynamically adjusting reasoning.

**Reasoning Evaluation.** Many studies evaluate reasoning process primarily via final-answer accuracy or related outcome metrics (Luo et al., 2025; Chern et al., 2023; Lu et al., 2023). However, recent works argue that ignoring intermediate steps can conceal logical flaws and inefficiencies such as redundant deliberation (Hammoud et al., 2025; Xia et al., 2025). Accordingly, step-level evaluation has been explored, including reference-based comparisons that align generated steps with gold solutions (Sawada et al., 2023). However, there are multiple valid reasoning paths that yield the same correct answer, limiting the fidelity of strict step matching. To improve scalability and coverage, recent efforts propose explicit evaluation rubrics and LLM-based proxies to judge reasoning quality along multiple criteria (Xia et al., 2025; Lee & Hockenmaier, 2025). Following this line, we use an LLM-based proxy to provide external feedback for large-scale experiments and complement it with human-in-the-loop evaluations when feasible to validate the effectiveness of CGI for efficient reasoning.

## C. Details of the Observation

### C.1. Experimental Setup for the Observations

**Setup.** We uniformly sample 100 questions of varying difficulty from MATH500 (Lightman et al., 2024). We use DeepSeek-R1-Distill-Qwen-32B (Shao et al., 2024) to generate full reasoning traces, and Qwen2.5-72B-Instruct (Yang et al., 2024a) as a judge for full outcomes and intermediate segments.

Our evidential chain proceeds in three steps. First, we quantify the lexical universality of candidate stage triggers via token-frequency analysis (Fig. 2a). Second, we examine whether these triggers align with semantic phase boundaries by mining their recurrent contextual patterns. Third, we test the *accuracy impact* of conjunction-triggered extensions by conditioning on whether the intermediate reasoning already contains a correct solution.

### C.2. Experimental Details for Observation 1

This section details the identification of prevalent lexical patterns that frequently co-occur with specific causal and transitional connectors. We systematically analyzed the textual context surrounding these connectors (designated as 'Base Word' in Table 8) to determine the most common associated words or phrases (listed under 'Patterns'). The regular expressions employed for this extraction are also presented.

#### C.2.1. THE CONTENT PATTERN

We use Qwen2.5-72B-Instruct to analyze the thinking behavior on problems of varying difficulty using the following instruction, which asks it to analyze the key conclusions in the thinking process and how they are connected.

**Details on counting the occurrences of target words.**

In our analysis, we count the frequency of individual tokens as produced by the tokenizer and then aggregate variants of the same word into a single lemma. For example, all of the following variants are grouped under the word "so":

$$\{" \ so", "So,", "So"\},$$

and similarly for "wait", "but", etc. Punctuation, capitalization, and leading spaces are normalized before aggregation.

In Figure 2, the horizontal axis represents each cluster of similar reasoning examples (i.e., all examples that share the same inference pattern). Within each cluster, we count how many times each word appears across all reasoning steps. We ignore purely punctuation tokens (e.g. ",", ".") and some stop-token variants that carry no semantic weight. After normalization, we

*Table 8.* Common content patterns for causal and transitional connectors. Co-occurrence words are words that appear near specific base words.

| Base Word | Co-occurrence words | Combined Method |
|---|---|---|
| Wait / wait | make sure, might, maybe, if, new, another, check, verify, whether, possible, hold on, not | `"Wait\s.{0,30}\s.+{co-words}"` `"wait\s.{0,30}\s.+{co-words}"` |
| But / but | if, new, another, check, verify, whether, maybe, possible, wait | `"But\s.{0,30}\s.+{co-words}"` `"but\s.{0,30}\s.+{co-words}"` |
| Alternative / alternative | if, new, another, check, verify, whether, maybe, possible, wait | `"Alternative\s.{0,30}\s.+{co-words}"` `"alternative\s.{0,30}\s.+{co-words}"` |
| So / so | can, is, would, be | `"So\s.{0,30}\s.+{co-words}"` `"so\s.{0,30}\s.+{co-words}"` |

keep only the top 100 words by total frequency.

**Raw rank and appearance counts for semantic tokens**: Among the top 150 tokens, most of them (90%) are non-semantic and those do not provide interpretable reasoning-structure information. From a semantic perspective, transitional and causal conjunctions are among the most frequent categories, compared with other semantic tokens such as "Let" and "check". Detailed raw ranks of semantic tokens at top-150 is in Table 9.

*Table 9.* Semantic Token appearance count with raw rank within top-150 for total 200 reasoning traces. We ignore the non-semantic tokens by "···".

| Rank | Token | Total Occurrences |
|---|---|---|
| ... | ... | ... |
| 21 | \u0120So | 2417 |
| ... | ... | ... |
| 39 | \u0120so | 2076 |
| ... | ... | ... |
| 42 | So | 1914 |
| ... | ... | ... |
| 52 | Wait | 1565 |
| ... | ... | ... |
| 64 | \u0120let | 1058 |
| ... | ... | ... |
| 66 | \u0120but | 1049 |
| ... | ... | ... |
| 74 | But | 970 |
| ... | ... | ... |
| 88 | \u0120Let | 798 |
| ... | ... | ... |
| 91 | Therefore | 729 |
| ... | ... | ... |
| 96 | \u0120think | 684 |
| 97 | maybe | 664 |
| ... | ... | ... |
| 100 | then | 657 |
| ... | ... | ... |
| 109 | Alternatively | 477 |
| ... | ... | ... |
| 126 | \u0120problem | 455 |
| ... | ... | ... |
| 129 | \u0120But | 339 |
| ... | ... | ... |
| 135 | \u0120Let | 256 |
| ... | ... | ... |
| 141 | \u0120check | 196 |
| ... | ... | ... |
| 150 | \u0120wait | 196 |

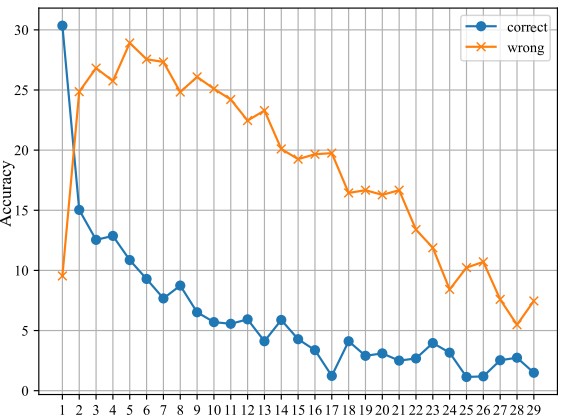 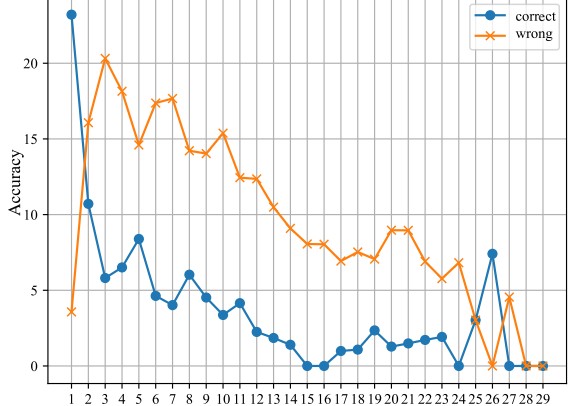

*(a)* The impact of extending thinking traces on final accuracy on the AIME24 dataset.

*(b)* The impact of extending thinking traces on final accuracy on the GPQA Diamond dataset.

*Figure 3.* Impact of intermediate thinking correctness on final accuracy on AIME24 and GPQA Diamond. The x-axis denotes the occurrence count of transitional conjunctions. The "correct" indicates that the reasoning prior to the current transitional conjunction contains the correct answer and yields a correct final prediction. The"wrong" indicates that the reasoning prior to the current transitional conjunction contains the correct answer but leads to an incorrect final prediction.

## C.3. The Experiment Details of Observation 2

### C.3.1. EXPERIMENTAL SETUP

To systematically investigate the relationship between reasoning dynamics and transitional conjunctions, we designed a rigorous pipeline involving trace generation, segmentation, and automated evaluation.

**Data Generation and Segmentation.** We prompted DeepSeek-R1-Distill-Qwen-32B to generate 16 reasoning traces for each question in our test set, utilizing a temperature of 1.0 and top-p of 0.95. To ensure a balanced analysis, we filtered the samples to retain questions that elicited both correct and incorrect final answers across the generated traces. Leveraging the transitional conjunctions identified in Observation 1, we segmented each reasoning trace into discrete sub-reasoning units.

**Automated Evaluation.** For each sub-reasoning segment preceding a conjunction, we employed Qwen2.5-72B-Instruct as an external judge. The judge was instructed to determine whether the reasoning trajectory up to that specific point already contained or implied the correct final answer, based on the ground truth and a reference solution. This evaluation allows us to compute the accuracy of the final answer conditioned on the state of intermediate sub-reasoning steps. The specific prompt template used for this evaluation is presented in Figure 4.

### C.3.2. MORE RESULTS AND ANALYSIS

Following the setup described above, we analyzed the impact of transitional conjunction usage on model performance. As initially observed in the MATH500 dataset (Figure 2c), the results demonstrate a distinct pattern: transitional conjunctions, when used judiciously, enhance accuracy by signaling necessary phases of self-correction or extended reasoning. However, an excessive accumulation of such conjunctions often correlates with a degradation in performance, indicative of "overthinking" or reasoning loops.

To test the generality of this observation, we extended our experiments to the AIME (Art of Problem Solving, 2024) and GPQA Diamond (Rein et al., 2024) datasets, maintaining the same experimental configuration. The results, illustrated in Figure 3a and Figure 3b, align consistently with our findings on MATH500. Across all datasets, we observe that appropriate prolongation of reasoning via transitional words improves outcomes, whereas overuse becomes detrimental. This cross-dataset consistency confirms that Observation 2 reflects a fundamental characteristic of the reasoning process in Large Reasoning Models.

---

**Evaluation Prompt for Subthinking**

**Task**
Your task is to analyze whether the current thinking is on the correct track based on the given corresponding question, standard solution, and correct answer.

**Standard**
There are two possible evaluation results and their corresponding criteria:

1. **Yes, the current thinking contains the right final answer:** The current thinking includes a standard solution or is close to or consistent with the standard solution, and the current thinking can obtain the correct answer in the future.

2. **No, the current thinking does not contain the right final answer:** The current thinking does not include the standard solution or is inconsistent with the standard solution, and the current thinking cannot obtain the correct answer in the future.

**Tips**

1. Your task is only to analyze whether the current thinking contains the right final answer or not, rather than giving the final answer.

2. Please do not provide any thinking or analytical process.

**Input:**
Given question: {question}
Standard solution: {solution}
Correct answer: {answer}
Current thinking: {current_thinking}

**Output:**
Your output must choose one of the two evaluation results, indicating whether the current thinking is on the correct track.
Your choice:

*Figure 4.* Evaluation Prompt for Subthinking

## D. Feedback Generation Criteria

To ensure consistency in evaluation, we employ an identical prompt for both human evaluators and the LLM proxy, as shown in Figure 5. This prompt guides the evaluators to assess the current state of reasoning and, if necessary, provide a revised reasoning state. Upon obtaining the judgment results, we populate the corresponding action-based feedback template. This structured feedback is then injected back into the reasoning loop to guide subsequent reasoning.

## E. Information-Theoretic Justification

This appendix provides a compact information-theoretic justification for why integrating feedback reduces uncertainty under our formulation, and summarizes the empirical validation.

### E.1. Notation and Definitions

Let:

- $T$: the model's target variable (e.g., next reasoning segment or final answer),

- $C$: the currently generated context,

- $F$: external information (feedback) provided during test-time interaction,

- $\theta$: model parameters,

- $H(\cdot \mid \cdot; \theta)$: (model-induced) conditional entropy,

- $I(\cdot; \cdot \mid \cdot; \theta)$: (model-induced) conditional mutual information.

All information-theoretic quantities are computed under the same joint distribution induced by the model (parameterized by $\theta$), ensuring standard identities apply.

**Proposition E.1** (Proposition 1). *For any valid $T, C, F$ (under a fixed $\theta$), conditioning on target-relevant feedback cannot increase uncertainty:*

$$H(T \mid C, F; \theta) \ \leq \ H(T \mid C; \theta). \tag{12}$$

*Proof.* We start from a fundamental result in information theory: conditional mutual information is non-negative:

$$I(T; F \mid C; \theta) \ \geq \ 0. \tag{13}$$

By the definition of conditional mutual information,

$$I(T; F \mid C; \theta) \ = \ H(T \mid C; \theta) \ - \ H(T \mid C, F; \theta). \tag{14}$$

Rearranging yields

$$H(T \mid C, F; \theta) \ = \ H(T \mid C; \theta) \ - \ I(T; F \mid C; \theta). \tag{15}$$

Since $I(T; F \mid C; \theta) \geq 0$, we obtain $H(T \mid C, F; \theta) \leq H(T \mid C; \theta)$, proving Proposition E.1. $\square$

### E.2. When Does Equality Hold?

Equality holds iff

$$I(T; F \mid C; \theta) = 0, \tag{16}$$

i.e., the feedback $F$ provides no additional information about $T$ once the context $C$ is given (feedback is target-irrelevant under the model-induced distribution).

---

**Prompt for Evaluator**

**Task**
Your task is to evaluate the provided thinking based on the given question and provide reasonable suggestions for improvement.

**Evaluation Criteria**
You need to evaluate whether the thinking is correct from the following aspects:

1. **Rationality:**

    - The current thinking satisfies the basic thinking rules. There are no situations such as skipping steps, missing steps, wrong steps, or incorrect thinking.
    - Reasonable knowledge. The knowledge involved in the current thinking conforms to common sense.
    - Reasonable solution. The current thinking uses a practical solution to solve the question.

2. **Completeness:**

    - Semantically draws the answer to the given question. There is an answer to the given question or a conclusion with the same efficiency as the answer.
    - Complies with the answer format. The thinking contains content that specifies the format of the answer, such as \boxed{answer}.

**Problem and Current thinking**
Given problem: {question}
Current inference result: {thinking}

**Tips**

1. Your task is only to evaluate the current thinking, not to provide the final answer to the question.

2. Please do not provide any thinking or analysis process unless it is necessary for the evaluation.

**Output Format**
Your analysis results must fall into one of four categories. Please answer strictly in the following format:

1. Rational and Complete. The current thinking is rational and contains the final answer, so it is complete.

2. Rational but Incomplete. The current thinking is rational but does not contain the final answer.

3. Irrational and Incomplete. The thinking is irrational and does not contain the final answer. The suggestion for improvement is:

4. Irrational but Complete. The thinking is irrational but contains the final answer. The suggestion for improvement is:

*Figure 5.* The Prompt for Evaluator

### E.3. Implication for CGI: Why Target-Relevant Feedback Helps

CGI intervenes in the reasoning process by using target-relevant feedback as informative side information that decreases the conditional entropy of the target. In CGI, feedback is generated with criteria oriented toward reaching the final answer (e.g., content-agnostic guidance such as Rationality and Completeness to adjust subsequent reasoning), which is designed to make $F$ informative about $T$ given $C$, thereby yielding $I(T; F \mid C; \theta) > 0$ in practice and reducing $H(T \mid C, F; \theta)$.

For convenience, define the uncertainty reduction as

$$\Delta H \triangleq H(T \mid C; \theta) - H(T \mid C, F; \theta) = I(T; F \mid C; \theta) \geq 0, \tag{17}$$

which makes explicit that the expected uncertainty reduction equals the conditional mutual information contributed by feedback.

### E.4. Empirical Verification

We empirically validate the predicted uncertainty reduction by estimating $H(T \mid C; \theta)$ and $H(T \mid C, F; \theta)$ before/after applying CGI across all datasets, and computing $\Delta H$ per instance. We observe that the proportion of cases with $\Delta H > 0$ reaches **98.73%**, supporting that the provided feedback in CGI is typically target-relevant and consistent with Proposition E.1.

## F. Implementation Details

### F.1. Implementation Details of baselines

**The Deployment Details of Model Baselines.** We evaluated several publicly available foundation models. For all foundation models listed below, inference was conducted with **temperature set to 0.6**, **top-p set to 1.0**, and utilized the **vLLM inference framework**:

- QwQ-32B

- Qwen2.5-72B-Instruct

- Qwen-Math-72B-instruct

- DeepSeek-R1-Distilled-Qwen2.5-7B

- DeepSeek-R1-Distilled-Qwen-14B

- DeepSeek-R1-Distilled-Qwen-32B

- ReasonFlux-7B

**The Deployment Details of L1** (Aggarwal & Welleck, 2025) L1 utilizes Length Controlled Policy Optimization (LCPO), a reinforcement learning technique, to train language models capable of adhering to specified generation length constraints during reasoning tasks. We specifically compare against the **L1-Max** variant, designed to control the maximum length of the generated output (e.g., Chain-of-Thought) while maintaining task accuracy. For our reproduction experiments using L1-Max: the maximum target token length ('ngold') was set to **8192**, including the input prompt; inference employed deterministic decoding with **temperature set to 0.6** and **top-p set to 1**; all computations were performed on **NVIDIA A800*4**.

**The Implementation of DEER.** DEER[2] is deployed via vLLM, first installing dependencies with pip install -r requirements.txt. It uses DeepSeek-R1-Distilled-Qwen2.5-32B as the core model, with max length=8192, temperature=0, and adjustable parameters (threshold 0.95, thinking ratio 0.6) before deployment via bash ./bashes/bash-vllm-deer.sh.

**The Implementation of SEAL.** SEAL[3] requires installing dependencies (pip install -r requirements.txt vllm=0.5.1 transformers=4.42.3) and setting up latex2sympy. It uses DeepSeek-R1-Distilled-Qwen2.5-32B on vLLM (max tokens=8192, temperature=0), with evaluation via evaluation/eval.sh after saving results.

---

[2] https://github.com/iie-ycx/DEER
[3] https://github.com/VITA-Group/SEAL

**The Implementation of SpecReason.** SpecReason[4] requires setting up a Python 3.12 environment, installing vLLM 0.8.2 and datasets, and applying a manual fix for speculative decoding in vLLM. It launches two vLLM servers (a 32B target model and a 1.5B draft model) and runs inference via `spec_reason.py` with a specified dataset, problem ID, token budget, and output directory to generate results.

**Length penalty only.** We fine-tune the target reasoning model **DeepSeek-R1-Distilled-Qwen2.5-7B** on the MATH train dataset using *only* the length-penalty objective in Equation 10 by **GRPO**. Training and evaluation are conducted under **8K** context window. We stop at **epoch 106** when the validation loss plateaus, and use the "length-penalty-only" model to directly get the answer across datasets.

**Prompt to think concisely.** At inference time (no additional training), we follow the concise thinking instruction (Nayab et al., 2024) to prompt the base model, **DeepSeek-R1-Distilled-Qwen2.5-7B** to minimize unnecessary intermediate steps and stop once the solution is determined, while preserving correctness with an 8k window size. The detailed prompt is stated in Figure 6.

---

**The Prompt for Thinking Concise**

{question} Think briefly about the answer.

---

*Figure 6.* The Prompt for Thinking Concise

### F.2. Implementation Details of CGI

This section provides detailed configurations and procedures for implementing the CGI.

- **Model training.** We build on **DeepSeek-R1-Distilled-Qwen2.5-7B** and start from extending its test vocabulary with `<reasoning_feedback>` and `</reasoning_feedback>` to accept external guidance. To ensure the efficiency of training and adaptation to new reasoning paradigm, A **LoRA** adapter is fine-tuned with **GRPO** on MATH train QA pairs. The Implementation of GRPO is based on the open source project[5] with a modified interactive reasoning sampling method. The details of LoRA are : $\alpha = 8$, $r = 8$, dropout$= 0.1$. GRPO: learning rate $1 \times e^{-4}$, LRM temperature 1.0, trajectories $= 8$ per question, batch size $= 1$, gradient accumulation $= 10$. We train until convergence using **PyTorch 2.6.0** (open-r1 implementation), with seed 42, max generation length **8192 tokens**, on $4\times$ **A800** GPUs.

- **Method deployment.** The LLM proxy is **Qwen2.5-72B-Instruct** (training-free), prompted with the predefined criteria. At test time, both the feedback proxy and the target reasoning model use temperature **0.6**. We cap the number of feedback rounds at **10**. Interventions are injected via the chat template by **appending the feedback enclosed by** `<reasoning_feedback>` and `</reasoning_feedback>` tags after the current partial reasoning in the assistant's turn, so the model continues generation conditioned on the updated context.

## G. The Details of Time and Computation Cost of Methods

We report comprehensive total inference time(s) and deployment GPU requirement (MB) for compared methods, confirming our framework's practical advantages. We apply vLLM to deploy models with fixed GPU usage. From Table 10, while the LLM proxy setting requires additional GPU memory (147456 MB) due to the local deployment of Qwen2.5-72B-Instruct for the controlled LLM proxy, this is not necessary for realistic deployment. In practice, the feedback service can be provided by an external API and the computational overhead is solely the target 7B reasoning model, requiring only 40,960 MB of GPU memory. CGI exhibits practical utility. From Table 10, CGI exhibits a small time increment (or even shorter time) compared to the base model, demonstrating its practicality.

### G.1. Elastic deployment requirements for CGI

Our method is designed for **elastic deployment**, accommodating various scenarios' requirements. The most resource-efficient configuration requires only the target model to be deployed locally. In Human-in-the-Loop deployment, the human

---

[4] https://github.com/ruipeterpan/specreason
[5] https://github.com/huggingface/open-r1

*Table 10.* Total time (s) and GPU memory (MB) of CGI and baselines across datasets. Minimum values are in **bold**, second minimum values are underlined.

| Method | MATH500 | | AIME24 | | GPQA Diamond | | LiveCodeBench | |
|---|---|---|---|---|---|---|---|---|
| | total_time(s) | GPU(MB) | total_time(s) | GPU(MB) | total_time(s) | GPU(MB) | total_time(s) | GPU(MB) |
| DeepSeek-R1-7B | 35.48 | **40960** | **49.54** | **40960** | 79.48 | **40960** | 72.51 | **40960** |
| ReasonFlux-7B | 71.89 | **40960** | 157.47 | **40960** | 126.92 | **40960** | 133.15 | **40960** |
| DEER | **21.75** | 65536 | 67.73 | 65536 | 47.26 | 65536 | 53.90 | 65536 |
| SEAL | 76.43 | **40960** | 247.40 | **40960** | 146.21 | **40960** | 254.14 | **40960** |
| SpecReason | 61.38 | 163842 | 250.09 | 163842 | 180.86 | 163842 | 208.46 | 163842 |
| CGI (LLM proxy) | 36.38 | 188416 | 62.27 | 188416 | 36.39 | 188416 | **39.92** | 188416 |
| CGI (human-in-the-loop) | 31.21 | **40960** | 55.48 | **40960** | **35.43** | **40960** | 59.85 | **40960** |

user naturally serves as the evaluator, eliminating the need for a separate LLM proxy. For large-scale automated testing, the evaluator role can be offloaded to external API-based models (e.g. Kimi-K2-Instruct). This minimal setup significantly lowers the barrier to entry by reducing hardware footprints and operational complexity.

For scenarios demanding lower latency and greater stability, co-locating the evaluator LLM with the target model is advantageous. While this requires additional computational resources, it offers minimal response times by eliminating network overhead from API calls and a stable and reproducible environment for large-scale batch experiments.

## H. The Details of Human-in-the-loop Experiment

### The setting details of Human-in-the-loop Experiment

Following prior human-evaluation practice that recruits graduate students as evaluators (Wang et al., 2023), we recruited ten graduate students with good English proficiency and diverse academic/professional backgrounds. We developed a dedicated user interface, as illustrated in Figure 7, to facilitate their interaction with CGI. During the experiment, evaluators provide feedback by selecting the appropriate options and can export the evaluation logs through the download function.

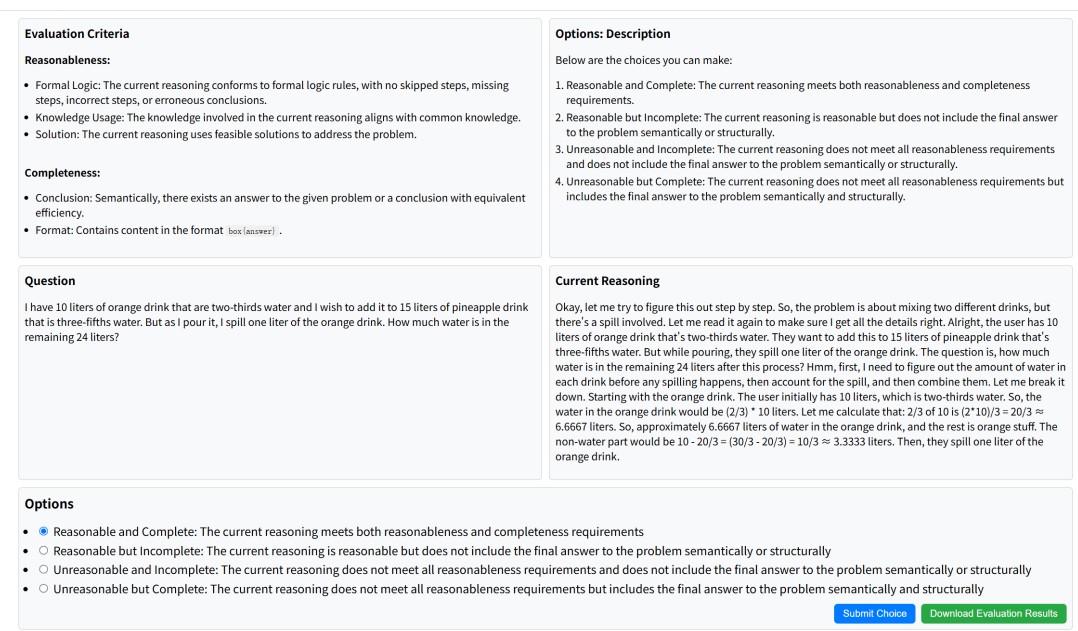

*Figure 7.* The interface for human evaluation

# I. The Interactive Interface for CGI

We have implemented a graphical interactive interface that is compatible with CGI reasoning paradigm for inserting external feedback as needed during the reasoning process: **1.** users **choose to have LLM proxy automatically** evaluate or **manually provide feedback** at the beginning of the reasoning stage, in Figure 8. **2.** The interactive interface **supports fine-grained reasoning feedback** based on multi-criteria and **optional suggestions for subsequent reasoning**, in Figure 9. **3.** The target model **generates </think> to terminate the reasoning** ( or be forced to insert </think> after maximum interactions) and provides the answer to the question, in Figure 10.

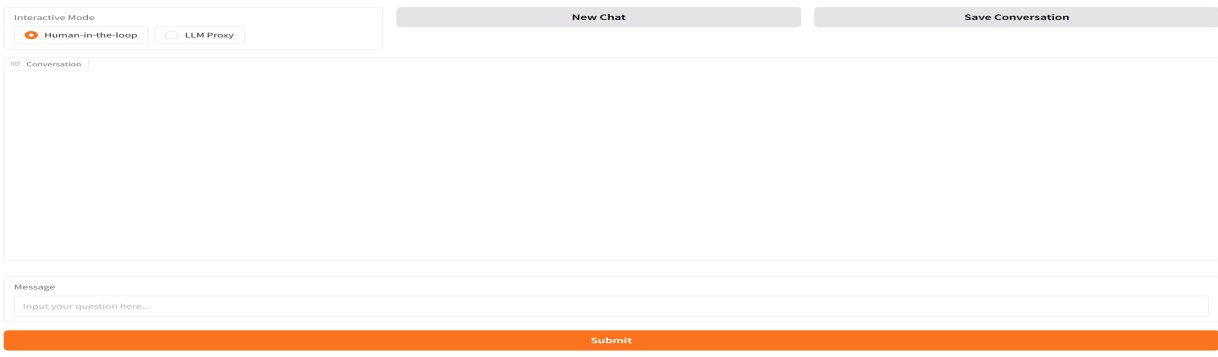

*Figure 8.* The Overview of Graphical Interface.

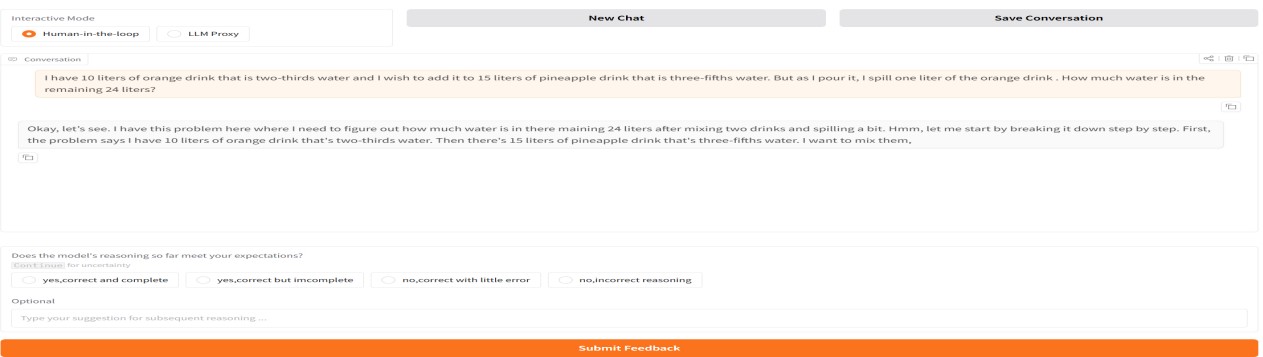

*Figure 9.* Four feedback options after halting the thinking process by specific transitional conjunctions.

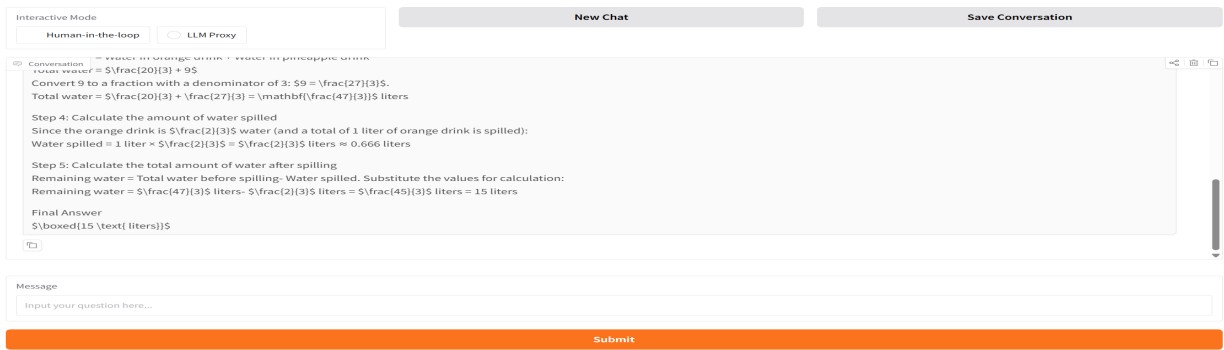

*Figure 10.* Get the final answer to the given questions after generating </think>.

## J. The Details of Ablation Study

In this section, we report the details of specific settings and key details in the ablation experiment.

### J.1. The details of ablation for specific intervention points

In this setting, we removed the model's ability to decide when to request feedback. Instead of using a special token to trigger an intervention, the reasoning process was automatically paused after generating a fixed length of 256 tokens. At this point, the generated text was evaluated by the LLM proxy, and feedback was provided for the model to continue its reasoning.

### J.2. The details of ablation for special tokens

For this ablation, we eliminated the use of special tokens `<reasoning_feedback>` and `</reasoning_feedback>`. The feedback generated by the LLM proxy was directly injected into the reasoning context as plain text, immediately following the model's last generated token.

### J.3. The details of ablation for trained model

To assess the impact of our training, this ablation study used the base target reasoning model without GRPO training in our composite reward function. In this setting, the original reasoning model, DeepSeek-R1-Distilled-Qwen2.5-7B, performs reasoning with the proposed CGI framework, receiving feedback from the LLM proxy.

### J.4. The details of ablation for Fine-Grained feedback

In this setting, the four distinct fine-grained feedback options were substituted with a simplified binary classification system. This system categorizes the reasoning process as either "Complete Thinking" or "Incomplete Thinking." The prompt used to elicit this binary evaluation and the corresponding expected output formats are detailed in Figure 11 and Figure 12. This comparative analysis helps isolate the impact of the nuanced guidance provided by our fine-grained approach versus a more basic feedback signal.

## K. Details of Additional Discussions

### K.1. The Details of different intervention strategies

We compared other intervention points under the same 8K context window in CGI: (1) intervening after every sentence, (2) intervening every 256 tokens, and (3) intervening at the \n\n token. To accommodate the high frequency of interventions in these strategies, the maximum number of interventions was increased to 1,000. All other experimental settings remained consistent.

## L. Case Study

### L.1. Potential applications in diverse tasks

We explore the effect of the proposed interactive reasoning mode in other critical (Wallace et al., 2024; Qin et al., 2024) and creative tasks (Wang et al., 2025) in Figure 13: with LLM proxy, we evaluate CGI on IFEval (Qin et al., 2024) for instruction following and on SEP (Wallace et al., 2024) for instruction hierarchy; with human-in-the-loop, we evaluate CGI on LitBench (Fein et al., 2026). These results indicate that CGI transfers across tasks and feedback sources.

#### L.1.1. EXPERIMENT DETAILS

**Evaluation on Instruction Following Task** We assess the interactive reasoning ability to adhere to instructions using the IFEval benchmark (Qin et al., 2024). The evaluation strictly follows the methodology proposed in the original paper. We compare the performance of the base model against our method CGI, where both are provided with a vanilla prompt with additional reminder "Think carefully about your decision", matching the interactive reasoning mechanism. In our setup, the interactive feedback is supplied by an LLM proxy. To quantify the capability, we report the accuracy, defined as the proportion of prompts for which the responses satisfy all verifiable instructions within the prompt.

---

**Evaluation Prompt for Subthinking**

**Task**

Your task is to evaluate the provided reasoning based on the given question and provide reasonable improvement suggestions.

**Evaluation Criteria**

1. *Rationality:*
– The reasoning follows basic logical rules (no skipped, missing, or incorrect steps).
– The knowledge used is consistent with common sense.
– The solution approach is practical and appropriate.

2. *Completeness:*
– The reasoning reaches a clear answer or conclusion equivalent to an answer.
– The reasoning specifies the answer format (e.g., contains "$\boxed{\text{answer}}$").

**Problem and Current Thinking**

Given problem: $\{question\}$
Current inference result: $\{reasoning\}$

**Tips**

1. Only evaluate the current reasoning; do not provide a new solution to the problem.
2. Avoid giving additional reasoning or analysis unless it is necessary for the evaluation.

**Output Format**

Your evaluation must fall into one of two categories (answer exactly as below):
1. `Complete Thinking`. The reasoning contains a valid final answer, so it is complete.
2. `Incomplete Thinking`. The reasoning does not contain the final answer to the question.

*Figure 11.* Evaluation Prompt for Subthinking

---

**Feedback Output Format**

**Complete**
`<reasoning_feedback>`
1. The current reasoning process is accurate and comprehensive, leaving no important details or logical gaps.
2. Generate the token `</think>` right away to halt any additional reasoning steps promptly.
3. Relying on the now-completed reasoning process that has been carried out, directly provide the final solution to the problem at hand.

`</reasoning_feedback>`
**Incomplete**
`<reasoning_feedback>`
1. The current reasoning is incomplete.
2. You need to continue reasoning to get the final answer. `</reasoning_feedback>`

*Figure 12.* Feedback Output Format

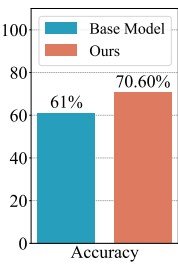

*(a)* Accuracy on IFEval.

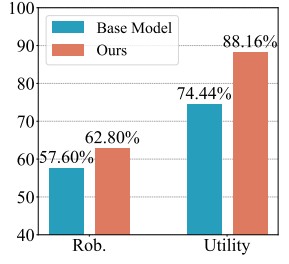

*(b)* Robustness and Utility on SEP.

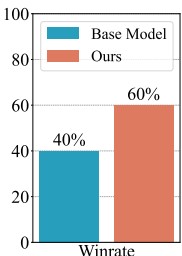

*(c)* Winrate on LitBench.

*Figure 13.* The effect of CGI by multi-source feedback in security and creative scenarios.

---

**Judgement prompt for Creative Task**

You are evaluating two creative writing responses (A and B) to the same writing prompt. These responses are similar to those posted on Reddit writing subreddits like r/WritingPrompts.

Your task is to predict which response would receive more upvotes from the Reddit community. Reddit users typically upvote creative writing that is engaging, original, well-written, and emotionally resonant.

When making your prediction, consider what makes content popular on Reddit:
- Originality and uniqueness of ideas
- Engaging narrative style and pacing
- Emotional impact and relatability
- Clever twists or satisfying conclusions
- Technical quality of writing

Story A:
{story_a}
Story B:
{story_b}
This is an experiment to test how well language models can predict human preferences in creative writing as expressed through Reddit's voting system.
Your verdict MUST follow this exact format:
Reasoning: [explain which response would likely get more Reddit upvotes and why]
Preferred: [A or B] (the one you predict would get more upvotes)

*Figure 14.* Judgement prompt for Creative Task

---

**Evaluation on Instruction Hierarchy Task** To evaluate the interactive reasoning capacity involving prioritized instructions, we use the SEP dataset (Zverev et al., 2025). This task challenges the model to execute a main, high-priority instruction while ignoring a secondary, low-priority one. Feedback during the reasoning process is provided by an LLM proxy. To match the interactive reasoning mechanism, we test the input with an additional reminder, "Think carefully about your creation" and measure performance using two metrics: $Rob.$ (robustness), which represents the proportion of low-priority instructions successfully ignored, and $Utility$, which quantifies the model's performance on the main task. The results in Figure 13b illustrate that CGI can effectively navigate conflicting instructions.

**Evaluation on Creative Task** We explore the application of CGI in creative domains through a story generation task, evaluated on the LitBench benchmark (Fein et al., 2026). Unlike the previous tasks, this setup incorporates human-in-the-loop feedback to guide the creative writing process. The evaluation is based on a preference comparison, where outputs from our method are judged against base model DeepSeek-R1-Distilled-Qwen2.5-7B with the additional requirement "Carefully consider your creation, I need creative content". Following the LitBench setup, we collected 20 samples, and the creative preferences were determined by an LLM-based evaluator (Claude-3.7-sonnet). The win rate, depicted in Figure 13c, highlights the potential of our interactive method in collaborative human-AI creative endeavors. The prompt used to judge is detailed in Figure 14.

## L.2. Reasoning Examples

This section presents a qualitative analysis through case studies to illustrate the practical advantages of our proposed method over the baseline model. We focus on two common undesirable behaviors observed in large language models: namely, "Overthinking" and "overshoot." "Overthinking" refers to scenarios where models engage in unnecessarily complex or prolonged reasoning processes before arriving at a conclusion. "overshoot" means that the model answered correctly initially, but then answered incorrectly during subsequent thinking. For each case, we will juxtapose the reasoning steps and final outputs generated by our method against those produced by the base model, thereby highlighting the improvements achieved.

### L.2.1. FAILURE CASE OF CGI

This case study analyzes a numerical reasoning task where low-quality evaluator feedback—which only addresses superficial expression errors and fails to identify core logical flaws—leads the reasoning agent to retain fundamental mistakes and produce an incorrect final answer. Below box illustrates a critical limitation of feedback-driven reasoning frameworks: the task outcome is heavily dependent on the quality of feedback, and incomplete or non-targeted feedback cannot guide effective error correction.

**failure case**

**User:**
Find the number of ordered pairs $(x, y)$, where both $x$ and $y$ are integers between $-100$ and $100$, inclusive, such that $12x^2 - xy - 6y^2 = 0$.

**Assistant:**

Okay, so I have this problem: I need to find the number of ordered pairs $(x, y)$ where both $x$ and $y$ are integers between $-100$ and $100$, inclusive. The condition given is $12x^2 - xy - 6y^2 = 0$. Hmm, let me try to figure this out step by step.

First, the equation is quadratic in both $x$ and $y$. Maybe I can factor it or find a relationship between $x$ and $y$. Let me write it down again: $12x^2 - xy - 6y^2 = 0$

Hmm, perhaps I can treat this as a quadratic equation in $x$. So, if I rewrite it: $12x^2 - xy - 6y^2 = 0$

Yes, treating it as a quadratic in $x$, the coefficients are: $A = 12$, $B = -y$, $C = -6y^2$

So, using the quadratic formula, $x$ would be: $x = \frac{y \pm \sqrt{y^2 + 4 \cdot 12 \cdot 6y^2}}{2 \cdot 12}$

Let me compute the discriminant: Discriminant $D = B^2 - 4AC = (-y)^2 - 4 \cdot 12 \cdot (-6y^2) = y^2 + 288y^2 = 289y^2$

Oh, that's nice because 289 is a perfect square, $17^2$. So $\sqrt{289y^2} = 17|y|$. Since $y$ is an integer, $|y|$ is non-negative, so $\sqrt{D} = 17|y|$.

So, plugging back into the quadratic formula: $x = \frac{y \pm 17|y|}{24}$

Hmm, now, let's consider two cases: $y$ positive and $y$ negative because of the absolute value.

Case 1: $y \geq 0$ Then $|y| = y$, so: $x = \frac{y + 17y}{24}$ So, two possibilities: $x = \frac{y + 17y}{24} = \frac{18y}{24} = \frac{3}{4}y$ Or $x = \frac{y - 17y}{24} = \frac{-16y}{24} = -\frac{2}{3}y$

Case 2: $y < 0$ Then $|y| = -y$, so: $x = \frac{y \pm 17(-y)}{24}$ So, again two possibilities: $x = \frac{y - 17y}{24} = \frac{-16y}{24} = -\frac{2}{3}y$ Or $x = \frac{y + 17(-y)}{24} = \frac{y - 17y}{24} = \frac{-16y}{24} = -\frac{2}{3}y$

**Evaluator:** 3. Correct reasoning. The current reasoning is on the right track with some expression error. Suggestions to point out the expression error: ==In the case where $y<0$, the second possibility should be $x = \frac{y + 17y}{24} = \frac{18y}{24} = \frac{3}{4}y$, not $x = \frac{y - 17y}{24} = \frac{-16y}{24} = -\frac{2}{3}y$.== Additionally, the reasoning should clarify that both $x = \frac{3}{4}y$ and $x = -\frac{2}{3}y$ need to be integers for valid solutions, which imposes constraints on the possible values of $y$.

**Assistant:**
Since $x$ and $y$ must be integers, let's analyze the two cases:

Case 1: $y \geq 0$
Subcase 1a: $x = \frac{3}{4}y$ For $x$ to be an integer, $y$ must be a multiple of 4. Let $y = 4k$ where $k$ is an integer. Then, $x = \frac{3}{4}(4k) = 3k$. So, $y = 4k$ and $x = 3k$.
Subcase 1b: $x = -\frac{2}{3}y$ For $x$ to be an integer, $y$ must be a multiple of 3. Let $y = 3m$ where $m$ is an integer. Then, $x = -\frac{2}{3}(3m) = -2m$. So, $y = 3m$ and $x = -2m$.
Case 2: $y < 0$
Subcase 2a: $x = \frac{3}{4}y$ Since $y$ is negative, $x$ would also be negative. For $x$ to be an integer, $y$ must be a multiple of 4. Let $y = -4k$ where $k$ is a positive integer. Then, $x = \frac{3}{4}(-4k) = -3k$. So, $y = -4k$ and $x = -3k$.
Subcase 2b: $x = -\frac{2}{3}y$ Similarly, $y$ is negative, so $x$ would be positive. For $x$ to be an integer, $y$ must be a multiple of 3. Let $y = -3m$ where $m$ is a positive integer. Then, $x = -\frac{2}{3}(-3m) = 2m$. So, $y = -3m$ and $x = 2m$. Now, we need to find all integer pairs $(x, y)$ within the range $-100 \leq x, y \leq 100$. Let's handle each subcase separately.

Subcase 1a: $y = 4k$ and $x = 3k$ $y = 4k$ must satisfy $-100 \leq 4k \leq 100$. Dividing by 4: $-25 \leq k \leq 25$. So, $k$ can take integer values from -25 to 25, inclusive. Therefore, there are $25 - (-25) + 1 = 51$ possible values of $k$. Each $k$ gives a unique pair $(x, y) = (3k, 4k)$.
Subcase 1b: $y = 3m$ and $x = -2m$ $y = 3m$ must satisfy $-100 \leq 3m \leq 100$. Dividing by 3: $\approx -33.333 \leq m \leq 33.333$. Since $m$ must be an integer, $m$ ranges from -33 to 33, inclusive. Therefore, there are $33 - (-33) + 1 = 67$ possible values of $m$. Each $m$ gives a unique pair $(x, y) = (-2m, 3m)$.
Subcase 2a: $y = -4k$ and $x = -3k$ $y = -4k$ must satisfy $-100 \leq -4k \leq 100$. Multiplying by -1 (reversing inequalities): $-100 \leq 4k \leq 100$. Dividing by 4: $-25 \leq k \leq 25$. So, $k$ can take integer values from -25 to 25, inclusive. Therefore, there are $25 - (-25) + 1 = 51$ possible values of $k$. Each $k$ gives a unique pair $(x, y) = (-3k, -4k)$.
Subcase 2b: $y = -3m$ and $x = 2m$ $y = -3m$ must satisfy $-100 \leq -3m \leq 100$. Multiplying by -1: $-100 \leq 3m \leq 100$. Dividing by 3: $\approx -33.333 \leq m \leq 33.333$. Since $m$ must be an integer, $m$ ranges from -33 to 33, inclusive. Therefore, there are $33 - (-33) + 1 = 67$ possible values of $m$.

**failure case**

Each $m$ gives a unique pair $(x, y) = (2m, -3m)$. Now, let's check for overlaps between these subcases. Overlaps occur when different subcases produce the same pair $(x, y)$. Looking at Subcase 1a and Subcase 2a:

Subcase 1a: $(3k, 4k)$

Subcase 2a: $(-3k, -4k)$

These are distinct unless $k = 0$, in which both give $(0, 0)$. Similarly, Subcase 1b and Subcase 2b:

Subcase 1b: $(-2m, 3m)$

Subcase 2b: $(2m, -3m)$

These are distinct unless $m = 0$, in which both give $(0, 0)$. Therefore, the only overlapping pair is $(0, 0)$, which is counted in all four subcases. So, when we sum up all subcases, we need to subtract the duplicates. Let's calculate the total number of pairs:

Subcase 1a: 51 pairs

Subcase 1b: 67 pairs

Subcase 2a: 51 pairs

Subcase 2b: 67 pairs

Total before considering overlap: $51 + 67 + 51 + 67 = 236$ Now, subtract the duplicates. Each of Subcase 1a and Subcase 2a includes $(0, 0)$ when $k = 0$. Similarly, Subcase 1b and Subcase 2b include $(0, 0)$ when $m = 0$.

So, $(0, 0)$ is counted four times in total, but it should be counted only once. Therefore, we need to subtract 3 instances of $(0, 0)$.

Hence, total number of ordered pairs is $236 - 3 = 233$.

**Evaluator:**

2. Correct reasoning. The current reasoning is practical and rational, but it is incomplete to generate the final answer, and further refine and complete the current reasoning.

**Assistant:** Calculating Total Pairs: Subcase 1a: 51 pairs Subcase 1b: 67 pairs Subcase 2a: 51 pairs Subcase 2b: 67 pairs Adding them up: $51 + 67 + 51 + 67 = 236$ However, we have overlaps: The pair $(0, 0)$ is counted four times (once in each subcase). Therefore, we need to subtract 3 duplicates to count it only once. So, total pairs $= 236 - 3 = 233$.

**Evaluator:**

3. Correct reasoning. The current reasoning is on the right track with some expression error. Suggestions to point out the expression error: In the final calculation, the pair $(0, 0)$ is counted four times, but it should be counted only once. Therefore, we need to subtract 3 instances of $(0, 0)$. However, the reasoning already correctly notes this and adjusts the total count from 236 to 233. The error is in the redundant reiteration of the calculation, which might confuse the reader. The final answer is correct, but the presentation could be more concise.

L.2.2. The Case for Mitigating the Overthinking Phenomenon

This subsection presents a case study specifically illustrating the phenomenon of "overthinking". The box below details the reasoning process and the resultant response generated by our proposed method, respectively. A key aspect of our approach is the integration of evaluator feedback, which allows the model to discern the precision of its thought process and subsequently make corrective adjustments. This mechanism is instrumental in mitigating "overthinking." For comparative purposes, the box below displays the base model's reasoning and its answer, which evidently exhibit the "overthinking" problem when addressing the user's query.

---

**The reasoning process of CGI for mitigating overthinking**

**User:**
What is $1^{(2^{235423523})}$?

**Assistant:**
Okay, so I need to figure out what $1^{(2^{235423523})}$ is. Hmm, let me break this down step by step. First, I remember that any number raised to any power is just multiplying that number by itself that many times. So, for example, $2^3 = 2 \times 2 \times 2 = 8$. But in this case, the base is 1, and the exponent is $2^{235423523}$. That exponent is a massive number, right? It's 2 raised to the power of 235,423,523. That must be an astronomically huge number. But

**Evaluator:**
4. Incorrect reasoning. The current reasoning completely does not contribute to the question, and a new reasoning is needed. Suggestions for exploring new reasoning: Focus on the properties of the number 1 when raised to any power, recognizing that $1^n = 1$ for any real number $n$. This will lead to the correct and immediate conclusion without the need to consider the magnitude of the exponent.

**Assistant:**
I know that 1 raised to any power is always 1. It's a fundamental rule in exponents. So, regardless of what the exponent is, as long as the base is 1, the result will always be 1.

Let me think about why that is the case. When you raise a number to a power, you're essentially multiplying that number by itself multiple times. For example, $1^2 = 1 \times 1 = 1$, $1^3 = 1 \times 1 \times 1 = 1$, and so on. No matter how many times you multiply 1 by itself, you'll always end up with 1.

So, even though the exponent here is a gigantic number like $2^{235423523}$, multiplying 1 by itself that many times doesn't change the outcome. It's still just 1.

I don't think there's any exception to this rule. Whether the exponent is zero, positive, negative, or even a fraction, as long as the base is 1, the result remains 1. For example, $1^0 = 1$, $1^{-1} = 1$, and $1^{1/2} = 1$. All of these still equal 1.

Therefore, applying this to the given problem, $1^{(2^{235423523})}$ must equal 1, no matter how large the exponent is.

I don't think I need to calculate the actual value of $2^{235423523}$ because it's unnecessary for solving this problem. The key insight is recognizing the base is 1, which simplifies everything.

So, in conclusion, the value of $1^{(2^{235423523})}$ is 1.

The value of $1^{(2^{235423523})}$ is 1 because any number raised to any power remains 1 if the base is 1.

$\boxed{1}$

---

**The reasoning process of baseline model**

**User:**
What is the value of

$$What\ is\ (1^{(2^{235423523})})?$$

**Assistant:**

Okay, so I need to figure out what $1^{(2^{235423523})}$ is. Hmm, let me break this down step by step.

First, I know that exponentiation is when you raise a number (the base) to a certain power (the exponent). In this case, the base is 1, and the exponent is $2^{235423523}$. That seems like a really huge exponent! But wait, the base is 1, so maybe that simplifies things?

Let me recall some basic exponent rules. I remember that any number raised to any power is just that number multiplied by itself that many times. So, for example, $2^3 = 2 \times 2 \times 2 = 8$. But here, the base is 1, so no matter how many times you multiply 1 by itself, you're always going to get 1.

Wait, so $1^2 = 1 \times 1 = 1$, $1^3 = 1 \times 1 \times 1 = 1$, and so on. So, regardless of the exponent, as long as the base is 1, the result is always 1. That makes sense because multiplying 1 by itself any number of times doesn't change its value.

So, applying that to this problem, $1^{(2^{235423523})}$ should just be 1, right? Because no matter how big the exponent is, it's still just 1 multiplied by itself that many times, which is still 1.

Let me double-check to make sure I'm not missing something. Is there any situation where 1 raised to a power isn't 1? Hmm, I know that in some cases, especially with limits in calculus, $1^\infty$ is considered an indeterminate form because it can approach different values depending on the context. But in regular exponentiation with finite numbers, $1^n$ is always 1, regardless of how large $n$ is.

So, in this case, $2^{235423523}$ is a finite number, albeit a very large one. Therefore, 1 raised to that power should still be 1.

Is there another way to look at this? Maybe using logarithms or something? Let me see. If I take the logarithm of $1^{(2^{235423523})}$, it would be $\log(1^{(2^{235423523})})$. But the logarithm of 1 is 0, so that would be $0 \times (2^{235423523})$, which is still 0. Then, exponentiating back, $e^0 = 1$. So that also gives me 1.

Another thought: maybe using properties of exponents. I know that $a^{b^c} = a^{(b^c)}$, which is the same as $(a^b)^c$, but actually, that's not the case. Wait, no, $a^{b^c}$ is different from $(a^b)^c$. The first one is $a$ raised to the power of $b^c$, whereas the second is $a^b$ raised to the power of $c$, which is $a^{b \times c}$. So, in our case, it's definitely 1 raised to the power of $2^{235423523}$, which is different from $(1^2)^{235423523}$. But regardless, since the base is 1, it's still 1.

Let me also think about the properties of exponents involving 1. I remember that 1 is the multiplicative identity, so multiplying by 1 doesn't change the value. Similarly, exponentiating 1 doesn't change the value. So, no matter how many times you exponentiate 1, it's still 1.

Another angle: if I consider the general case, $1^n$ for any integer $n$, positive, negative, or zero. Well, except for $n = 0$, but even $1^0 = 1$ by definition. So, in all cases, it's 1. Therefore, even with such a massive exponent, it doesn't change the outcome.

Wait, could there be a case where $1^{\text{something}}$ isn't 1? For example, in modular arithmetic? Hmm, in modular arithmetic, sometimes things wrap around, but unless specified, regular exponentiation is just 1. So, unless we're working modulo something, which isn't indicated here, it should still be 1.

Also, thinking about the definition of exponentiation, $a^b$ is the product of $b$ copies of $a$ multiplied together. So, if $a = 1$, then no matter how many copies you have, multiplying them all together is still 1. So, that's another confirmation.

Is there a way this could be something else? Maybe if the exponent is zero? Wait, $2^{235423523}$ is definitely not zero. 2 raised to any positive integer is at least 2, so $2^{235423523}$ is a huge number, but not zero. So, $1^{(\text{that huge number})}$ is still 1.

Wait, what if the exponent was zero? Then it would be $1^0 = 1$. But since the exponent here is $2^{235423523}$, which is definitely not zero, so that doesn't apply.

Also, in some contexts, like in limits, as I mentioned before, $1^\infty$ is indeterminate, but here it's a finite exponent, so it's just 1.

Another thought: maybe if the exponent was a variable, but in this case, it's a specific number, so that's not an issue.

**base model thinking and answer–Overthinking(Continue)**

Wait, could I write this as $1^{2^{235423523}} = (1^2)^{235423523}$? But no, exponentiation is right-associative, so $a^{b^c} = a^{(b^c)}$ and not $(a^b)^c$. So, in this case, it's 1 raised to $2^{235423523}$, not $(1^2)^{235423523}$. But since both are 1, it still ends up as 1.

Wait, let me clarify the associativity. So, exponentiation is right-associative, meaning that $a^{b^c}$ is evaluated as $a^{(b^c)}$ and not $(a^b)^c$. So, in our case, it's definitely 1 raised to $2^{235423523}$, which is 1, and not $(1^2)^{235423523}$, which is also 1. So, in either case, it's 1, but the right-associative property just tells us the order of operations.

So, even if we considered the other way, it still results in 1. So, that doesn't change the outcome.

Is there a way this could be something other than 1? I can't think of any. All the exponent rules I know point to it being 1.

Wait, let me consider the definition of exponentiation. For any integer $a$, $a^0 = 1$ if $a \neq 0$, and $a^1 = a$. So, for $a = 1$, $1^n$ is 1 for any integer $n$. Since $2^{235423523}$ is an integer, positive, it's still 1.

Is the exponent allowed to be that large? I mean, in regular exponentiation, the exponent can be any integer, whether small or large. So, even if it's a huge number, 1 raised to it is still 1.

Wait, could there be some kind of overflow or something in computer science? But since this is a mathematical problem, not a computational one, we don't have to worry about that.

Another thought: if I use logarithms, as I did earlier, $\log(1^{(2^{235423523})}) = 0 \times (2^{235423523}) = 0$, and then $e^0 = 1$. So that also gives me 1.

Alternatively, using natural logarithm: $\ln(1^{(2^{235423523})}) = (2^{235423523}) \times \ln(1) = 0$, so exponentiating gives $e^0 = 1$. Same result.

Is there any other mathematical operation or function that might change this? For example, if I take the factorial of 1? But no, that's not relevant here. Or maybe the square root or something? The square root of 1 is 1, but that's not the operation we're dealing with here.

Wait, another angle: in some contexts, especially in abstract algebra, 1 can be the multiplicative identity in different structures. But regardless of the structure, multiplying 1 by itself any number of times will still give 1. So, even in more abstract settings, $1^n = 1$.

Also, thinking about complex numbers, 1 is still 1. Even if you raise 1 to any power in the complex plane, it's still 1. So, no change there.

Wait, maybe I'm overcomplicating this. Let me step back. The problem is $1^{(2^{235423523})}$. The base is 1, exponent is $2^{235423523}$. Since 1 raised to any power is 1, regardless of how big the exponent is, the result is 1.

Therefore, after considering all these angles, I can't find any reason why this wouldn't be 1. It seems pretty straightforward. Therefore, the Final Answer: is $\boxed{1}$.

L.2.3. THE CASE FOR MITIGATING OVERSHOOT

This subsection presents a case study specifically illustrating the phenomenon of "overshoot," where a model continues processing despite having already reached a correct internal answer. The content below details the reasoning process and the resultant response generated by our proposed method, respectively. A crucial feature of our method is its enhanced ability to recognize when a sufficient answer has been formulated internally, prompting a more timely termination of the redundant reasoning process. This prevents the model from unnecessarily elaborating on or potentially altering an already adequate solution, thereby effectively addressing "overshoot." In contrast, the content below showcases the base model's reasoning and response, which clearly demonstrate the "overshoot" tendency: even after an appropriate answer is likely identified internally, the model continues its deliberation, leading to a less optimal final output.

---

**The reasoning process of CGI for mitigating overshoot**

**User:**
Each week, between 30 and 50 students show up for an archery class run by Betty and Wilma. Usually the students break up into groups of equal size for target practice. However, this week, Betty noticed that she could not break the students up into multiple groups of equal size. Wilma noticed that if she and Betty both joined the students in practicing, they still could not break the archers up into groups of equal size. How many students showed up to the archery class this week?

**Assistant:**

Okay, so I have this problem about Betty and Wilma's archery class. Each week, between 30 and 50 students show up. Usually, they break up into groups of equal size for target practice. But this week, Betty noticed she couldn't break them into multiple groups of equal size. Then Wilma noticed that even if she and Betty joined, they still couldn't break them up into equal groups. I need to figure out how many students showed up this week. Alright, let me break this down. So, normally, the number of students can be divided into groups of equal size. That means the number of students is divisible by some number greater than 1, right? So, the number of students must be a composite number, not a prime number. But this week, Betty couldn't break them into multiple groups of equal size. That suggests that the number of students this week is actually a prime number because prime numbers can't be divided into equal groups other than 1 and themselves. So, the number of students is prime. But

**Evaluator:**
Rational but incomplete. The current reasoning is practical and rational, but it does not contain the final answer to the question. So it is correct but incomplete.

**Assistant:**
So, Betty couldn't split them into multiple groups, implying the number is prime. Then Wilma noticed that even if both she and Betty joined, making the total number of students $N + 2$, they still couldn't split them into equal groups. So, $N + 2$ is also prime. Therefore, we're looking for a prime number $N$ between 30 and 50 such that $N + 2$ is also prime. That means $N$ and $N + 2$ are twin primes.
Now, let's list the prime numbers between 30 and 50:
31, 37, 41, 43, 47
Now, let's check which of these primes have another prime two greater than them:

- 31: $31 + 2 = 33$, which is not prime (divisible by 3)

- 37: $37 + 2 = 39$, which is not prime (divisible by 3)

- 41: $41 + 2 = 43$, which is prime

- 43: $43 + 2 = 45$, which is not prime (divisible by 5)

- 47: $47 + 2 = 49$, which is not prime (7x7)

So, the only pair where both $N$ and $N + 2$ are prime is 41 and 43.
Therefore, the number of students this week must be $\boxed{41}$.

---

**The reasoning process of base model**

**User:**
Each week, between 30 and 50 students show up for an archery class run by Betty and Wilma. Usually the students break up into groups of equal size for target practice. However, this week, Betty noticed that she could not break the students up into multiple groups of equal size. Wilma noticed that if she and Betty both joined the students in practicing, they still could not break the archers up into groups of equal size. How many students showed up to the archery class this week?

**Assistant:**
Okay, so I need to figure out how many students showed up to the archery class this week. Let's see. The problem says that usually, the students break up into groups of equal size, but this week Betty noticed she couldn't do that. Then Wilma said that even if she and Betty joined, they still couldn't break them up into equal groups.

Hmm. So, let me break this down. Normally, the number of students is between 30 and 50. Let me denote the number of students as N. So, N is between 30 and 50.

Now, the students usually break up into groups of equal size. That means N is a number that can be divided into equal groups, so N must be a composite number because it has factors other than 1 and itself. But this week, Betty noticed that she couldn't break them into multiple groups of equal size. So, that means that N doesn't have any divisors other than 1 and itself? Wait, no, that would make N a prime number, but if N is prime, then the only way to break them into groups is one group or N groups, but the problem says they usually break into multiple groups, meaning more than one group but less than N. So, if N is prime, they can't break them into multiple groups, which is what Betty noticed this week.

But wait, the problem says usually they can break into groups of equal size, so N is composite. But this week, Betty noticed that she couldn't break them into multiple groups, implying that N is prime? That seems contradictory because if N is prime, usually they can break into groups, but this week they can't. Maybe I need to think differently.

Wait, no. Maybe the usual grouping is into groups of a certain size, say, groups of 2, 3, 4, etc. So, if N is a number that is composite, it can be divided into equal groups, but this week, for some reason, Betty couldn't. Maybe the number of groups she was thinking of is such that it's not a divisor of N. So, Betty was trying to divide them into a specific number of groups, which didn't work, but it's not necessarily that N is prime. It could be that N is composite, but Betty was trying to split them into a group size that doesn't divide N.

Wait, let me re-read the problem: "However, this week, Betty noticed that she could not break the students up into multiple groups of equal size." So, Betty noticed that she couldn't break them into multiple groups of equal size. So, maybe she tried to split them into, say, 2 groups or 3 groups, and it didn't work. So, the number of students is such that it can't be divided into equal groups of 2, 3, etc., but usually, they can. Hmm, so maybe N is a number that is prime? Because if N is prime, you can't split them into groups of 2, 3, etc., without having a remainder.

But wait, usually, they can break up into groups, so N is composite. So, this week, Betty noticed that she couldn't break them into multiple groups, so maybe she tried to split them into groups of a certain size, but that size doesn't divide N. But it's not necessarily that N is prime, because N could be composite but have some divisors that don't work.

Wait, maybe the key is that N is a prime number. Because if N is prime, then you can't split them into multiple equal groups unless you have 1 group or N groups. But the problem says that usually, they break into multiple groups, so N is composite. Hmm, but this week, Betty noticed that she couldn't break them into multiple groups, so maybe the number of students is such that it's prime? But that contradicts the usual situation. Wait, maybe I need to think about the problem differently.

Let me parse the problem again: "However, this week, Betty noticed that she could not break the students up into multiple groups of equal size." So, Betty tried to split them into multiple groups, but she couldn't. So, the number of students, N, is a number such that it cannot be divided into equal groups of a certain size. But usually, they can be divided into equal groups, so N is composite. So, this week, Betty tried to divide them into, say, 2 groups, but N is odd, so she couldn't. Alternatively, she tried to split them into 3 groups, but N isn't a multiple of 3. Hmm.

## The reasoning process of base model (Continue)

But then, Wilma noticed that if she and Betty both joined the students in practicing, they still couldn't break the archers up into groups of equal size. So, when adding 2 more students (Betty and Wilma), the new number of students is N + 2, and this number also cannot be divided into equal groups of any size. So, both N and N + 2 are numbers that cannot be divided into equal groups of multiple sizes, meaning both are prime numbers? Wait, if N is composite, but N + 2 is prime, that's possible. Or both N and N + 2 are composite but cannot be divided into equal groups? Hmm, that seems conflicting because if N is composite, it can be divided into equal groups of certain sizes, unless those specific group sizes don't divide N.

Wait, maybe N is a prime number? Because if N is prime, you can't split them into multiple groups, but if N is prime, usually they can't split them either. But the problem says that usually, they can break into groups, so N is composite. Hmm, this is confusing.

Wait, let me think of it another way. If N is such that it's composite, but it's also such that N + 2 is prime. So, Betty couldn't split N into multiple groups, but N + 2 is prime, so even if Betty and Wilma join, they still can't split into multiple groups. So, maybe N is a composite number where N is one less than a prime number. So, N is composite, N + 2 is prime.

But let's test this idea. Let's list numbers between 30 and 50, and see which are composite, and then check if N + 2 is prime.

Let me list the numbers from 30 to 50:

30, 31, 32, 33, 34, 35, 36, 37, 38, 39, 40, 41, 42, 43, 44, 45, 46, 47, 48, 49, 50.

Now, composite numbers are all except the primes: 31, 37, 41, 43, 47.

So, composite numbers between 30 and 50: 30, 32, 33, 34, 35, 36, 38, 39, 40, 42, 44, 45, 46, 48, 49, 50.

Now, for each composite number N, check if N + 2 is prime.

Let's go one by one:

30: 30 + 2 = 32, which is composite. So, N + 2 is composite. Not good.

32: 32 + 2 = 34, composite. Not good.

33: 33 + 2 = 35, composite. Not good.

34: 34 + 2 = 36, composite. Not good.

35: 35 + 2 = 37, which is prime. Okay, so 35 is a candidate.

36: 36 + 2 = 38, composite. Not good.

38: 38 + 2 = 40, composite. Not good.

39: 39 + 2 = 41, which is prime. Okay, 39 is another candidate.

40: 40 + 2 = 42, composite. Not good.

42: 42 + 2 = 44, composite. Not good.

44: 44 + 2 = 46, composite. Not good.

45: 45 + 2 = 47, which is prime. Okay, 45 is another candidate.

46: 46 + 2 = 48, composite. Not good.

48: 48 + 2 = 50, composite. Not good.

49: 49 + 2 = 51, composite. Not good.

50: 50 + 2 = 52, composite. Not good.

So, the composite numbers N in 30-50 where N + 2 is prime are 35, 39, and 45.

Now, let's check these numbers.

First, N = 35. 35 is 5 * 7, so it's composite. N + 2 = 37, which is prime. So, Betty couldn't split 35 into multiple equal groups because 35 is composite but maybe she tried to split it into groups that don't divide 35, but wait, 35 can be divided into 5 groups of 7 or 7 groups of 5. So, actually, she should be able to split them, but the problem says she couldn't. Hmm, that seems contradictory.

Wait, maybe Betty tried to split them into 2 groups, but 35 is odd, so you can't split into 2 equal groups. Similarly, 35 is not divisible by 3: 35 / 3 is about 11.666, so not an integer. So, maybe Betty tried to split them into groups of 2 or 3, which didn't work, so she couldn't. So, N = 35 is a possible answer.

Similarly, N = 39. 39 is 3 * 13. So, you can split them into 3 groups of 13 or 13 groups of 3. So, usually, they can split into multiple groups. But this week, Betty couldn't.

## The Reasoning Process of Base Model (Continue)

So, perhaps Betty tried to split them into 2 groups, but 39 is odd, so can't split into 2 equal groups. Similarly, 39 is not divisible by 4: 39 / 4 is 9.75, so not integer. So, maybe she tried to split into 2 or 4 groups, which didn't work. So, N = 39 is another possible answer.

Similarly, N = 45. 45 is 9 * 5, so it can be split into 5 groups of 9 or 9 groups of 5. So, usually, they can split into multiple groups. But this week, Betty couldn't. Maybe she tried to split into 2 groups, which is not possible because 45 is odd. Similarly, 45 isn't divisible by 4: 45 / 4 = 11.25, so no. So, N = 45 is another candidate.

So, we have three candidates: 35, 39, and 45.

Now, the problem says that this week, Betty couldn't break them into multiple groups, and Wilma noticed that if they both joined, they still couldn't break them into equal groups. So, N is such that N is composite, and N + 2 is prime. But we have multiple possibilities. How do we choose between 35, 39, and 45?

Wait, let's think about the initial condition: "However, this week, Betty noticed that she could not break the students up into multiple groups of equal size."

So, if Betty couldn't break them into multiple groups, it means that N is a composite number, but she couldn't split them into a specific number of groups, say, k groups, where k is a divisor of N. But N is composite, so it has some divisors other than 1 and itself. So, if N is composite, it can be split into multiple groups, but Betty couldn't.

Wait, maybe Betty tried to split them into a prime number of groups, but N is not a multiple of that prime. For example, if she tried to split them into 2 groups, which is prime, but N is odd, so it can't be split into 2 equal groups. But then, if N is even, you can split into 2 groups, so she could have done that. So, N must be odd, because she couldn't split into 2 groups. So, N is odd.

Looking back at our candidates: 35, 39, 45 are all odd, so that's consistent.

Now, let's see if N is 35, 39, or 45, which one is the correct answer.

Wait, maybe we can think of another condition. The problem says that when Betty and Wilma join, so N + 2, they still can't break the archers into equal groups. So, N + 2 is prime, as we saw earlier.

But also, N must be such that N is composite, but N + 2 is prime.

So, in this case, all our candidates satisfy that N is composite, N + 2 is prime.

But perhaps there's another condition: that N is such that it can't be split into multiple groups because it's a prime power? Hmm, not necessarily.

Wait, let me think differently. Maybe N is a composite number, but N is a semiprime, meaning it's the product of two primes, so it can only be split into two groups of equal size or into groups of size equal to the prime factors.

So, if N is a semiprime, it can be split into two equal groups or into groups equal to its prime factors.

But if Betty couldn't split them into multiple groups, maybe she tried to split them into more than two groups, but N is a semiprime, so it can't be split into more than two equal groups unless those groups are size 1 or N.

Wait, but semiprimes can sometimes be split into more groups if the number of groups is a divisor. For example, 15 is 3 * 5, so it can be split into 3 groups of 5 or 5 groups of 3. So, if N is a semiprime, it can be split into two equal groups or into groups equal to the prime factors.

So, if Betty tried to split them into 3 groups, but N isn't a multiple of 3, then she couldn't. So, if N is a semiprime, and 3 doesn't divide N, then Betty couldn't split them into 3 groups.

Similarly, if N is 35, which is 5 * 7, she couldn't split into 3 groups because 35 isn't a multiple of 3.

Similarly, N = 39 is 3 * 13, so she couldn't split into 5 groups because 39 isn't a multiple of 5.

Similarly, N = 45 is 9 * 5, which is $3^2 * 5$, so she couldn't split into 7 groups or something else.

Wait, maybe the key is that N is such that it's a semiprime where the two prime factors are distinct and greater than 2, so that N is odd and can't be split into 2 groups, and also, N + 2 is prime.

But I think all three numbers 35, 39, 45 satisfy that.

Wait, maybe the problem is that N is a composite number where N is not a multiple of 2, 3, or 5, but N + 2 is prime.

## The reasoning process of base model (Continue)

Wait, 35 is 5 * 7, so it's a multiple of 5. 39 is 3 * 13, so it's a multiple of 3. 45 is 9 * 5, which is $3^2 * 5$.

So, perhaps N is a composite number that is a multiple of 3 or 5, but N + 2 is prime. So, 35, 39, and 45 all fit that. Hmm.

Alternatively, maybe the problem is that N is such that it's a composite number, and N + 2 is a prime, but N is also a perfect square or something.

Wait, 35 is not a perfect square, 39 is not, 45 is not.

Alternatively, maybe N is a composite number such that N is one less than a prime, but N + 2 is prime. Wait, no, N + 2 is prime in our case.

Wait, maybe I should look for N such that N is composite, N + 2 is prime, and N is the smallest such number in the range 30-50. But 35 is the first one.

Wait, let's check N = 35: 35 is composite, 35 + 2 = 37 is prime. So, that fits.

But let's see if N = 35, can Betty split them into multiple groups? 35 can be split into 5 groups of 7 or 7 groups of 5. So, if Betty tried to split them into 5 groups, she could, but if she tried to split them into 3 groups, she couldn't. So, if Betty couldn't split them, maybe she tried to split them into 3 groups, which doesn't work because 35 isn't divisible by 3.

Similarly, N = 39: 39 can be split into 3 groups of 13 or 13 groups of 3. So, if Betty tried to split them into 5 groups, which 39 isn't divisible by, she couldn't.

N = 45: 45 can be split into 5 groups of 9 or 9 groups of 5. So, if Betty tried to split them into 7 groups, which 45 isn't divisible by, she couldn't.

So, all three numbers fit the conditions.

Wait, so how do we choose between them? The problem says that this week, Betty couldn't break them into multiple groups of equal size. So, N is composite, but Betty couldn't split them, meaning that N is composite, but N isn't divisible by the number of groups she tried to split them into.

But since the problem doesn't specify what group size Betty was trying, maybe we need another condition.

Wait, maybe N is such that it's composite, but N is also a prime power? But 35 is 5*7, 39 is 3*13, 45 is $3^2 * 5$. So, they're all composite with multiple prime factors.

Wait, maybe N is such that N is a multiple of a prime, but N + 2 is prime. So, N is a multiple of 5 or 3 or something, and N + 2 is prime.

But all three numbers are multiples of 3 or 5.

Wait, maybe N is the only composite number in the range where N + 2 is prime. But no, all three are.

Alternatively, perhaps N is such that N is a Carmichael number or something, but that might be overcomplicating.

Wait, maybe I need to think about the fact that when Betty and Wilma join, making N + 2, which is prime, so they still can't split them into equal groups. So, N + 2 is prime, so they can't split them into multiple groups, which is consistent.

But N is composite, so it can be split into equal groups, but Betty couldn't split them into equal groups, so maybe she tried to split them into a specific number of groups that didn't divide N.

Wait, but all three numbers N = 35, 39, 45 can be split into multiple equal groups, except when Betty tried to split them into groups that don't divide N.

But the problem doesn't specify what group size Betty was trying, so maybe all three numbers satisfy the condition.

But the problem is asking for a specific number, so I must be missing something.

Wait, let's reread the problem:

"How many students showed up to the archery class this week?"

Betty noticed that she couldn't break the students up into multiple groups of equal size.

### The reasoning process of base model (Continue)

Wilma noticed that if she and Betty both joined the students in practicing, they still couldn't break the archers up into groups of equal size.

So, perhaps the key is that N is composite, so Betty could have broken them into equal groups, but she couldn't. That suggests that N is composite, but Betty was trying to split them into a prime number of groups, which didn't divide N. But N is composite, so it must have some prime factors. So, if Betty tried to split them into a prime number of groups, but N isn't divisible by that prime, then she couldn't.

So, for example, if Betty tried to split them into 2 groups, but N is odd, so she couldn't. Similarly, if she tried to split them into 3 groups, but N isn't a multiple of 3.

So, in our case, N is either 35, 39, or 45.

Now, let's check if N is 35: 35 is 5 * 7, so it can be split into 5 groups of 7 or 7 groups of 5. So, if Betty tried to split them into 2, 3, 4, 6, etc., groups, she couldn't. So, she couldn't break them into multiple groups of equal size. So, that works.

Similarly, for N = 39: 39 is 3 * 13. So, Betty couldn't split them into 2, 4, 5, etc., groups, but she could into 3 or 13. So, if she tried to split them into 2 groups, she couldn't. So, that works too.

N = 45: 45 is 9 * 5, so it's $3^2 * 5$. So, Betty couldn't split them into 2, 4, 7, etc., groups, but she could into 3, 5, 9, or 15. So, if she tried to split them into 2 groups, she couldn't. So, that works too.

So, all three numbers fit.

Wait, maybe the problem is that N is such that N is composite, but N is also a perfect square or something.

But 35 is not a perfect square, 39 isn't, 45 isn't.

Alternatively, maybe N is a multiple of 3 and 5, like 45, so it's a multiple of 15. But 35 is a multiple of 5 and 7, and 39 is a multiple of 3 and 13.

Alternatively, maybe N is the only number in that range where N is composite, and N + 2 is prime, but also N is such that N is the product of two primes, both greater than 2.

So, 35 is 5*7, both primes greater than 2. 39 is 3*13, 3 is a prime, but 3 is not greater than 2 in terms of being odd, but it's still a prime. 45 is $3^2 * 5$, so it's not a product of two distinct primes.

Wait, so 35 is the product of two distinct primes, both greater than 2. 39 is the product of two distinct primes, one of which is 3. 45 is the product of a prime squared and another prime.

So, perhaps the answer is 35, because it's the only one where both prime factors are greater than 2. Hmm, but that might be a stretch.

Alternatively, maybe the problem is expecting N to be a number such that N is composite, N + 2 is prime, and N is the smallest such number in the range.

But 35 is the first one in the range 30-50.

Wait, let me think of another way. Maybe N is a composite number where N is not a multiple of 2, 3, or 5, but that seems restrictive.

Wait, 35 is 5*7, so it's a multiple of 5. 39 is 3*13, multiple of 3. 45 is 9*5, multiple of 3 and 5.

Wait, so they all are multiples of 3 or 5. So, maybe all three numbers are possible.

But the problem is expecting a specific answer, so maybe I need to think differently.

Wait, let me check the initial conditions again.

Betty couldn't split them into multiple groups of equal size. So, N is composite, so Betty should have been able to split them, but she couldn't. So, maybe she tried to split them into a number of groups equal to a prime, but N is not a multiple of that prime.

Similarly, when Wilma joins, N + 2 is prime, so they still can't split them into multiple groups.

But is there another condition?

Wait, the key might be that N is such that it's composite, but N is also a prime power. Wait, 35 is not a prime power, 39 isn't, 45 isn't.

Alternatively, maybe N is a Carmichael number, but 35 is 5*7, which is not a Carmichael number because 5 - 1 doesn't divide 35 - 1, which is 34. 5 - 1 = 4, 34 divided by 4 is 8.5, which is not an integer. Similarly, 7 - 1 = 6, 34 divided by 6 is about 5.666, not integer. So, 35 isn't a Carmichael number.

## The Reasoning Process of Base Model (Continue)

Similarly, 39: 3 - 1 = 2, 38 divided by 2 is 19, which is integer. 13 - 1 = 12, 38 divided by 12 is about 3.166, not integer. So, 39 isn't a Carmichael number either.

45: 3 - 1 = 2, 44 divided by 2 is 22, integer. 5 - 1 = 4, 44 divided by 4 is 11, integer. So, 45 is a Carmichael number? Wait, 45 is $3^2 * 5$. For a number to be a Carmichael number, it must be square-free and satisfy Korselt's criterion, which is that for all prime divisors p of n, p - 1 divides n - 1.

So, 45 is not square-free because it's $3^2 * 5$. So, it's not square-free, so it can't be a Carmichael number. So, that idea doesn't hold.

Hmm. Maybe I need to think of another approach.

Wait, let me consider the initial statement: usually, they can break into groups of equal size, but this week they couldn't. So, N is composite, but this week, Betty couldn't. So, maybe N is a composite number, but it's a prime power, like $2^5 = 32$, which is composite, but it can only be split into 2 groups of 16 or 16 groups of 2. So, if Betty tried to split them into 3 groups, she couldn't. So, N = 32 is a possibility.

But 32 is in the range 30-50, and 32 + 2 = 34, which is composite. So, does N = 32 fit?

So, N = 32 is $2^5$, composite. Betty couldn't split them into multiple groups, maybe 3 groups, since 32 isn't divisible by 3. And when Wilma joins, N becomes 34, which is composite, so they still can't split into equal groups because 34 can be split into 2 groups of 17 or 17 groups of 2, but 17 is prime, so they can't split them into more than 2 groups unless they try to split into 17 groups, which is impractical.

Wait, but 34 is 2 * 17, so they can split into 2 groups of 17 or 17 groups of 2. So, if Betty and Wilma join, making 34, they can split into 2 groups of 17 or 17 groups of 2. So, they can split them, which contradicts the problem statement that they still couldn't break them into equal groups.

So, N = 32 is out because when they add 2, N + 2 = 34 is composite, but they can split them into equal groups.

So, N + 2 must be prime for them to still not be able to split them into equal groups. So, N + 2 must be prime, so N + 2 is prime. So, N must be such that N is composite, and N + 2 is prime.

So, that brings us back to 35, 39, 45.

So, all three numbers satisfy N composite, N + 2 prime.

But the problem is asking for a specific number. So, maybe I need to think of another condition.

Wait, let me think about the fact that N is such that it's composite, and it's the product of two primes, both of which are greater than 2. So, 35 is 5 * 7, both primes greater than 2. 39 is 3 * 13, 3 is a prime, but 3 is not greater than 2 in a meaningful way, but still, it's a prime. 45 is $3^2 * 5$, so it's not a product of two distinct primes.

So, maybe 35 is the answer because it's the product of two primes, both greater than 2, so it can't be split into 2 groups, and when adding 2, it becomes 37, which is prime, so can't be split into multiple groups.

Alternatively, maybe 39 is the answer because it's the product of 3 and 13, both primes, and when adding 2, it becomes 41, which is prime.

==Wait, 39 + 2 is 41, which is prime.==

Similarly, 35 + 2 is 37, which is prime.

45 + 2 is 47, which is prime.

So, all three numbers fit.

Wait, maybe the key is that N is the only composite number in that range where N is composite, and N + 2 is prime, and N is a semiprime.

But 35, 39, and 45 are all semiprimes or products of primes with powers.

Wait, maybe the problem is expecting N to be 35 because 35 is the first such number in the range where N is composite, N + 2 is prime, and N is a product of two distinct primes, both greater than 2.

Alternatively, maybe the problem is expecting the answer 35, 39, or 45, but it's expecting one.

Wait, maybe I need to think of another angle.

## The reasoning process of base model (Continue)

Wait, the problem says that usually, they can break into groups of equal size, so N is composite, but this week, Betty couldn't. So, N is composite, but Betty couldn't split into multiple groups, which might mean that N is a prime power, like $2^5 = 32, 3^3 = 27$, but 27 is below 30.

Wait, 32 is in the range, 32 is composite, but it's a prime power. So, can you split 32 into multiple equal groups? Yes, into 2 groups of 16, 4 groups of 8, 8 groups of 4, 16 groups of 2. So, Betty could split them into multiple groups, but the problem says she couldn't. So, that contradicts.

So, N is composite, but Betty couldn't split them into equal groups, which suggests that N is composite, but it's not a multiple of some number that Betty tried to split them into. So, N is composite, but Betty tried a prime number of groups, which didn't divide N.

So, for example, if N is 35, Betty couldn't split into 3 groups because 3 doesn't divide 35.

If N is 39, Betty couldn't split into 5 groups because 5 doesn't divide 39.

If N is 45, Betty couldn't split into 7 groups because 7 doesn't divide 45.

So, all three fit.

Wait, maybe the problem is that N is such that N is composite, N + 2 is prime, and N is square-free.

35 is square-free, 39 is square-free, 45 is not square-free because it's 9 * 5. So, 45 is not square-free.

So, maybe N is square-free. Then, 35 and 39 are square-free, 45 isn't.

So, perhaps N is 35 or 39.

But the problem is expecting a specific number.

Wait, maybe 35 is the answer because it's the smallest composite number in the range where N + 2 is prime.

But 35 is not the smallest, 30 is composite, but 30 + 2 = 32, which is composite. So, 30 doesn't fit.

Similarly, 32 doesn't fit because N + 2 is composite.

So, 35 is the first composite number where N + 2 is prime.

But 35 is 5 * 7, so it's square-free.

Similarly, 39 is 3 * 13, square-free.

So, both are square-free.

Hmm, maybe both are possible answers, but the problem is expecting one.

Wait, let me check the original problem again.

"How many students showed up to the archery class this week?

Betty noticed that she could not break the students up into multiple groups of equal size.

Wilma noticed that if she and Betty both joined the students in practicing, they still could not break the archers up into groups of equal size. How many students showed up to the archery class this week?"

Wait, so the problem is presented as a single answer. So, perhaps I need to think of another condition.

Wait, maybe N is such that N is composite, N + 2 is prime, and N is a multiple of 3 or 5.

But all three numbers 35, 39, 45 are multiples of 3 or 5.

Alternatively, maybe N is such that N is the product of two primes, both of which are greater than 2, making N square-free and odd.

So, 35 is 5 * 7, both primes greater than 2.

39 is 3 * 13, 3 is not greater than 2, but still, it's a prime.

45 is $3^2 * 5$, not square-free.

So, 35 is the only one where both prime factors are greater than 2.

Therefore, N = 35.

So, I think the answer is 35.

Final Answer The number of students who showed up to the archery class this week is $\boxed{35}$.

