# OpenReview forum: "Intervene When It Doubts: Conjunction-Guided Interactive Reasoning"
_ICML.cc/2026/Conference — ICML 2026 regular_

### Official Review · Reviewer_MsGR · 2026-02-24

**Soundness:** 2
**Presentation:** 3
**Significance:** 3
**Originality:** 3
**Overall Recommendation:** 4
**Confidence:** 2

**Summary:**

The paper proposes "Conjunction-Guided Intervention" (CGI), a reasoning framework designed to mitigate "overthinking" and "overshoot" in Large Reasoning Models (LRMs). The authors' core contribution is the empirical discovery that transitional conjunctions (e.g., "but," "wait," "alternatively") serve as linguistic markers of "doubt," where models often trigger unnecessary or self-destructive reasoning extensions. CGI leverages these markers as intervention points, pausing the generation to inject external, state-based feedback regarding the "rationality" and "completeness" of the current thinking trace. This feedback is provided by either a human or an LLM proxy such as Qwen2.5-72B model. To ensure the target model effectively adapts to this interactive paradigm, the authors employ GRPO with LoRA. The framework aims to optimize the trade-off between reasoning accuracy and token length across challenging benchmarks like MATH, AIME, and GPQA.

**Compliance With Llm Reviewing Policy:**

Affirmed.

**Final Justification:**

I have raised my score to weak accept according to the convincing results provided by the authors.

**Key Questions For Authors:**

- I'm not sure but the paper emphasizes "efficiency" by showing reduced reasoning lengths for the 7B target model, while the authors use a Qwen2.5-72B-Instruct proxy as the evaluator. If we account for the total FLOPs consumed by both the 7B model and the 72B proxy across multiple intervention rounds, is the system actually more efficient than simply letting the 7B model think for longer, or even using a 14B/32B model directly? Is there are a "Total System Compute" comparison?
- The performance gains are most visible in Mathematics (AIME/MATH). In these domains, "Rationality" and "Completeness" are relatively objective. How would this framework function in open-ended or subjective reasoning tasks (e.g., legal analysis or ethical reasoning)? Does the task-agnostic nature of your criteria lead to a "generic" reasoning style that might overlook domain-specific nuances? I think there should be at least some quantative results demostrating this.
- I notice that the ablation study (Table 4) shows a massive performance drop when using the "w/ base model" setting. Why does the un-tuned base model (DeepSeek-R1-7B) fail so catastrophically when presented with external feedback? Is it because the model perceives the injected feedback as a "distraction" that breaks its internal CoT (Chain of Thought) coherence, or is it simply a failure to recognize the special <reasoning_feedback> tags?
-  Why is the GRPO work based on your results? If SFT was performed on "synthesized traces," did the model simply learn to mimic the length of those traces rather than the logic of the feedback? How exactly did GRPO help the model learn the "stop" signal more effectively than SFT?

**Limitations:**

Yes

**Strengths And Weaknesses:**

The paper’s greatest strength lies in its intuitive and well-grounded observation that conjunctions serve as semantic "checkpoints" for reasoning depth. While previous work has looked at internal numerical signals (like entropy) to stop reasoning, using the model’s own natural language transitions is a novel and highly interpretable approach. The authors provide strong empirical evidence (Observation 1 and 2) to justify why these specific points are critical for intervention. Furthermore, the integration of GRPO training specifically to improve "feedback responsiveness" is a modern and technically sound choice that aligns with the current state-of-the-art in RL for reasoning. The results are impressive, demonstrating significant token savings (e.g., reducing reasoning length by thousands of tokens) while simultaneously improving accuracy, particularly on logic-heavy tasks like AIME where overthinking often leads to "overshoot" errors.
For the weaknesses,
- The "hidden" computational and latency cost of the intervention process itself. While the paper claims to improve reasoning efficiency, the reliance on a much larger LLM proxy (a 72B model judging a 7B model) suggests that the total system-wide FLOPs might actually increase; the "token savings" in the 7B model are achieved at the expense of multiple high-latency API calls or local inferences from a massive evaluator model. This makes the "efficiency" claim somewhat narrow, as it ignores the overhead of the judge.
- The framework introduces significant real-world latency; pausing generation up to 10 times per query to wait for external feedback makes it impractical for time-sensitive applications
- The evaluation is heavily skewed toward mathematics and coding; as noted in the appendix, the gains are weaker in areas where intermediate reasoning artifacts are less verifiable, suggesting the framework’s "task-agnostic" feedback criteria may struggle with subjective or creative domains where "rationality" is harder for a proxy to judge accurately.

---

> ### Author Rebuttal · Authors · 2026-03-31
>
> We sincerely thank the reviewer for recognizing the **novel and highly interpretable approach**, **strong empirical evidence**, and the **strong accuracy-length trade-off**.
>
> >Q1: What is "efficiency" considering the evaluator overhead? Do you provide total system compute comparison?
>
> **A1.** The reasoning efficiency for CGI is to reduce the redundancy in reasoning while maintaining the task performance. CGI has advantages in system-wise token generation and time latency.
> * **Efficiency** is jointly balancing **task accuracy** and **token generation** by **reducing redundant reasoning** from overthinking and overshoot.
> * **CGI has advantages in system-wise computing.** From Table I, CGI yields the lowest reasoning token generation with clear time latency advantages on challenging reasoning tasks.
>
> Table I. Average total system compute of CGI across reasoning tasks, including #Tokens (the average number of tokens within one reasoning trace) and total_time (time latency in second） that cover both reasoning and feedback.
> ||#Tokens| total_time|
> |-|-|-|
> |Base model(7B)|4141.38|61.75|
> |CGI (Ours)|**1638.75**|**43.74**|
>
> >Q2: Pausing generation up to 10 times per query to wait for external feedback makes it impractical for time-sensitive applications.
>
> **A2.** CGI does not introduce prohibitive latency in its target settings and provides unique value in reasoning controllability and interpretability:
> * **CGI uses far fewer than 10 interactions.** the average number of interventions is below **6.29** across datasets and 10 is the force-stop upper bound.
> * **The interaction overhead is offset by shorter reasoning.** From Tab. I, although CGI pauses for external feedback, it reduces redundant reasoning and remains competitive in end-to-end latency.
>
> >Q3: How well does CGI generalize to open-ended or subjective tasks, where the task-agnostic criteria may miss domain-specific nuances?
>
> **A3.** CGI generalizes beyond math and coding to open-ended and subjective tasks.
> * **Task-agnostic intervention principle.** CGI suggests that state-based intervention **without solution hints** can transfer to various reasoning benchmarks.
> * **Concrete Criteria need domain adaptation.** While preserving the same design principle, concrete criteria should be adjusted for new domains without solution hints.
> * **CGI offers benefits beyond efficiency.** Tab. II provide preliminary transfer evidence for CGI on **safety-related** and **creative** tasks.
>
> Table II. The results of CGI on safety-related and creative tasks.
> |  |SEP||LitBench|
> |-|-|-|-|
> ||robustness|utility|preference rate|
> |Base model|57.60|74.44|0.40|
> |**CGI (Ours)**|**62.80**|**88.16**|**0.60**|
>
> >Q4: Why does the untuned base model fail in CGI from Table 4?
>
> **A4.** The untuned 7B base model appears to be **less stable under the interactive reasoning paradigm**, likely because it is not well adapted to reliably interpret and follow external feedback.
> * **Interactive mismatch.**  The base model is trained for uninterrupted self-generated CoT, not for reasoning with external feedback inserted mid-generation. Therefore, it may not naturally incorporate the injected feedback as actionable guidance.
> * **Limited feedback-following ability.** Without such adaptation, the model does not consistently utilize injected feedback, which can disrupt continuation and contribute to the large drop in Tab. 4. From Table III, removing the feedback tags yields only limited accuracy improvement (43.56→48.27), suggesting that the issue is not mainly formatting, but rather limited feedback-following ability under the interactive paradigm.
>
> Table III. Average performance of the base model in CGI with or without feedback tags.
> ||accuracy|#Tokens|
> |-|-|-
> |base model|43.56|1715.72|
> |base model (without feedback tags)|48.27|1742.99|
>
> >Q5: Why is the GRPO work in this work? Why did SFT fail in CGI? How exactly did GRPO help the model learn the "stop" signal more effectively than SFT?
>
> **A5** SFT imitates surface patterns from synthesized traces, while GRPO further optimizes the stop-or-continue decision under the interactive paradigm:
> * **GRPO works** by optimizing correctness, format, and length rewards, enabling the model to adapt to feedback while directly targeting the dual core task objectives of answer accuracy and reasoning efficiency under the proposed interactive reasoning paradigm.
> * **SFT fails** due to exposure bias—it only mimics static response patterns without learning to condition on dynamic feedback.
> * **GRPO achieves better stop-signal learning.** GRPO yields a higher self-termination rate than SFT, showing the model learns to emit </think> proactively rather than relying on forced stops.
>
> Table IV. The self-termination rate (%) of CGI with SFT vs. GRPO.
> ||MATH500|AIME 24|GPQA Diamond|LiveCode Bench|
> |-|-|-|-|-|
> |w/ SFT|78.67|53.50|56.71|44.13|
> |**w/ GRPO**|93.96|76.67|79.90|74.85|

---

> > ### Author Rebuttal · Reviewer_MsGR · 2026-04-03
> >
> > I thank the authors for their rebuttal. I will raise my score according to the convincing results provided by the authors.

---

> > > ### Author Response · Authors · 2026-04-03
> > >
> > > Thank you for the careful follow-up and for your active engagement throughout the rebuttal process. We sincerely appreciate your thoughtful comments. Although our initial rebuttal has addressed your concerns as you have chosen "(a) Fully resolved", we now provide the **dataset-level results** of those we reported the **average result** in the initial rebuttal for completeness:
> > >
> > > * **Dataset-level result of Table I（end-to-end reasoning token and time latency）** We supplement the corresponding dataset-level results of Table I in Table A. These results further support that CGI maintains competitive system-level cost, with clearer advantages on more challenging reasoning tasks such as GPQA and LiveCodeBench.
> > >
> > > Table A. Total system compute of CGI across reasoning tasks, including the generated token **(#Token.)** and time latency (**total_time.**) that cover both **reasoning** and **feedback**. Base model is the untuned DeepSeek-R1-Distilled-7B and we apply the tuned DeepSeek-R1-Distilled-7B through GRPO in CGI.
> > >
> > > | Method       | MATH500                     |                  | AIME24                     |                  | GPQA Diamond               |                  | LiveCodeBench               |                  |
> > > |-|-|-|-|-|-|-|-|-|
> > > |              | #Token. | total_time. | #Token.   | total_time. | #Token.    | total_time. | #Token.   | total_time. |
> > > | Base model  | 1864.33                    | 35.48            | 3374.50                    | **49.54**        | 5020.86                    | 79.48            | 6305.82                    | 72.51            |
> > > | CGI (Ours)   | **1228.30**                | 36.38            | **2387.40**                | 62.27            | **1378.65**                | **36.39**        | **1560.64**                | **39.92**        |
> > >
> > > * **More results of Table II** We supplement the preliminary transfer evidence for CGI on **safety-related** and **creative** tasks in Table B, where IFEvl and SEP are safety-related benchmarks and LitBench is the creative writing task.
> > >
> > > Table B. Quantitative results of CGI on safety-related and creative open-ended tasks.  Further details have been reported in Appendix H.2 of the submission.
> > > |            | IFEvl       |  SEP    |   |      | LitBench            |
> > > |------------|-------------|------|--------|------|---------------------|
> > > |            | accuracy | robustness | utility |      | Preference winrate |
> > > | Base model | 61.00        | 57.60 | 74.44  |      | 0.40                 |
> > > | **CGI (Ours)**       | **70.60**        | **62.80** | **88.16**  |      | **0.60**                 |
> > >
> > > * **The base model is the untuned DeepSeek-R1-Distilled-7B and was conducted on the representative datasets in Table III.** The dataset-level result in Table C confirms that removing the feedback tags yields limited accuracy improvement, suggesting that the inefficiency issue is not mainly a formatting issue, but rather the limited feedback-following ability of the small untuned model under the interactive paradigm.
> > >
> > > Table C. Performance of the untuned model in CGI with or without feedback in no tags.
> > > |                          | MATH500 | MATH500 | AIME24| AIME24| GPQA Diamond| GPQA Diamond | LiveCode Bench | |
> > > |--------------------------|------------------|----------------|-----------------|---------------|-----------------------|--------------------|------------------------|-----------------------|
> > > |              |accuracy   |  #Token. | accuracy    | #Token. |accuracy     |  #Token.  |accuracy   |  #Token. |
> > > | base model                 | 75.23            | 1190.85        | 33.75           | 2151.13       | 37.24                 | 2011.89            | 28.00                  | 1509.00               |
> > > | base model (without feedback tags)| 85.40      | 1205.35        | 40.73           | 2032.12       | 39.29                 | 2115.02            | 28.00                  | 1619.50               |
> > >
> > >
> > > ----
> > > We hope the above dataset-level result is helpful for you to update your detailed scores to reflect your revised assessment. Thank you again for your time and kind consideration.

---

### Official Review · Reviewer_wKHn · 2026-03-02

**Soundness:** 2
**Presentation:** 3
**Significance:** 2
**Originality:** 2
**Overall Recommendation:** 4
**Confidence:** 4

**Summary:**

This work addresses the phenomena of redundant generation and the unintended alteration of correct answers in Large Reasoning Models caused by self-doubt during complex logical tasks. To this end, it proposes an interactive reasoning framework named CGI (Conjunction-Guided Interactive Reasoning), which identifies natural conjunctions within the reasoning chain as intervention anchors to dynamically introduce external feedback for guiding the model's generation.

**Compliance With Llm Reviewing Policy:**

Affirmed.

**Final Justification:**

The authors addressed my concerns during the rebuttal period and provided additional experiments, which made the work more complete. Therefore, I have revised my score from 3 to 4.

**Key Questions For Authors:**

1. Compared to the high-cost methods mentioned in the paper, what specific advantages does CGI offer in terms of training data volume, training time, and other related metrics?

**Limitations:**

Yes.

**Strengths And Weaknesses:**

Strengths:
1. The paper provides a detailed discussion on how redundant conjunctions influence the model's reasoning outcomes, offering insightful observations into the mechanisms behind overthinking and overshooting.
2. The proposed method enables the 7B-scale model to achieve improved accuracy while effectively reducing the output length.

Weaknesses:
1. Through the frequency analysis in Figure 2(a), the paper attempts to demonstrate that conjunctions are among the "most frequent" categories in reasoning traces. However, the chart lacks a baseline comparison and does not show the distribution of non-conjunction high-frequency words for contrast.
2. The paper primarily demonstrates the effectiveness of CGI on a 7B-scale model, but it lacks experimental results showing the framework's performance across models of different scales or magnitudes.
3. The framework relies significantly on the quality of the intervention model itself. A weaker intervention model would result in a corresponding reduction in performance improvements.
4. The paper points out that existing retraining methods involve high computational costs, which motivates the proposal of CGI. However, the CGI framework itself involves a fine-tuning process based on the GRPO algorithm, which is typically also computationally expensive.

Minor remarks (do not affect the score):
1. In the "Implementation Details" section, the authors mention that the backbone model is DeepSeek-R1-Distilled-Qwen2.5-7B. According to the official DeepSeek release records and the context of the paper, this appears to be a typo and should be DeepSeek-R1-Distill-Qwen-7B.

---

> ### Author Rebuttal · Authors · 2026-03-31
>
> We sincerely thank the reviewer for recognizing the strengths of our work, including the **insightful observation** and **strong efficiency gains of CGI**.
>
> >Q1: The chart of Figure 2(a) lacks a baseline comparison and does not show the distribution of non-conjunction high-frequency words for contrast.
>
> **A1.** Figure 2(a) focuses on high-frequency **semantic tokens** rather than raw top-ranked tokens, and we will **add baseline comparisons** in the revision. The details are as follows:
> * **Figure 2(a) focuses on high-frequency semantic tokens** instead of all raw top-ranked tokens. From Table I, most **raw high-frequency tokens** are **non-semantic** and those do not provide interpretable reasoning-structure information. From a **semantic perspective**, transitional and causal conjunctions are among the most frequent categories, compared with semantic tokens such as “Let” and “check”.
> * **Revise Figure 2(a) with baseline comparison in the revision.** We will add design details and baseline comparisons in the revision to make this observation more transparent, while the **core observation remains unchanged**.
>
> Table I. Raw statistics of high-frequency semantic tokens in MATH500 (partial listing due to space constraints). Non-semantic tokens are omitted with ellipses (…). The complete statistics are added to Appendix C.2 in the revision.
> |Rank|Token|Average frequency in one trace|
> |-|-|-|
> |1|\u0120|55.89|
> |...|...|...|
> |21|\u0120so|4.83|
> |23|\u0120so|4.15|
> |28|So|3.28|
> |33|Wait|3.13|
> |...|...|...|
> |53|\u0120but|2.10|
> |55|But|1.94|
> |58|Therefore|1.45|
> |...|...|...|
> |89|\u0120maybe|1.35|
> |93|Alternatively|1.30|
> |96|\u0120think|1.28|
> |...|...|...|
> |100|\u0120then|1.03|
> |109|\u0120Let|1.01|
> |126|\u0120check|0.98|
> |129|\u0120wait|0.98|
> |...|...|...|
> |134|Maybe|0.95|
> |...|...|...|
>
> >Q2:The paper primarily demonstrates the effectiveness of CGI on a 7B-scale model, but it lacks experimental results showing the framework's performance across models of different scales or magnitudes.
>
> **A2.** Experiments show that CGI consistently improves reasoning efficiency on both **DeepSeek-R1-Distilled-7B** and **QwQ-32B**. The details are as follows:
> * **CGI generalizes across models.** From Table I, the intervention framework of CGI works on **QwQ-32B** for reasoning efficiency, which is not limited to the **7B model**.
> * **The intervention mechanism of CGI works for efficient reasoning.** The core **intervention mechanism** of CGI, including intervention points and feedback criteria, is **general across models** to adaptively **reduce redundant reasoning**.
>
> Table I. Average result of CGI on QwQ-32B across datasets. #Tokens denotes the average number of tokens within one reasoning trace.
> ||accuracy|#Tokens|
> |-|-|-|
> |QwQ-32B|55.31|4621.00|
> |CGI(QwQ-32B)|57.67|2565.22|
>
>
> >Q3: The framework relies significantly on the quality of the intervention model itself. A weaker intervention model would result in a corresponding reduction in performance improvements.
>
> **A3.** Experiments show that **CGI supports proxies of different scales**, since the feedback is generated based on **task-agnostic criteria**. The details are as follows:
> * **CGI supports proxies of different scales.** From Table III, CGI with different proxies consistently outperforms the base model.
> * **CGI is the reasoning intervention framework rather than a mechanism for knowledge transfer.** The proxy serves as a judge and provides feedback using **task-agnostic criteria** on the current reasoning state, **without solution hints**.
>
> Table III. Performance of CGI with proxies of different scales. CGI maintains a favorable accuracy-length trade-off through task-agnostic criteria on the reasoning state.
> ||accuracy|#Tokens|
> |-|-|-|
> |Base model|42.15|4141.38|
> |CGI with 7B proxy|54.20|1305.63|
> |CGI with 14B proxy|56.99|1294.35|
> |CGI with 32B proxy|59.59|1282.36|
> |CGI with 72B proxy|66.13|1162.35|
>
> >Q4: Compared to the high-cost methods mentioned in the paper, what specific advantages does CGI offer in terms of training data volume, training time, and other related metrics?
>
> **A4.** From Table IV, CGI achieves **lower training cost** in terms of **finetuning parameters**, **training data** and **compute**. GRPO is conducted on the **lightweight LoRA** for feedback responsiveness. We freeze the backbone parameters and optimize only the LoRA module.
>
> Table IV. Training cost comparison between CGI and baselines.
> ||finetuning parameters|training data volume|training context|Training algorithm|GPU cost|Iterations|
> |-|-|-|-|-|-|-|
> |L1-Max|1.5B|40k|4k|LCPO|8\*A100|300 steps|
> |ReasonFlux-7B|7B|15k+10k|8k|HRL+DPO|8\*A100|6 + 6（epochs）|
> |**CGI（Ours）**|**1.26M**|**7.5k**|**8k**|**GRPO**|**4\*A800**|**120 steps**|

---

> > ### Author Rebuttal · Reviewer_wKHn · 2026-04-02
> >
> > Thank you for the rebuttal.
> > 1. My concern is specifically that Figure 2(a) does not provide comparisons against high-frequency non-conjunction tokens, which makes it hard to assess the claim that causal and transitional conjunctions are “among the most frequent tokens in reasoning.”
> > In addition, in Table I, do all omitted entries marked by “...” correspond to non-semantic tokens? If so, the table can help show that semantic non-conjunction tokens tend to rank lower and resolve my concern;
> > 2. For the additional CGI(QwQ-32B) evidence, could the authors provide dataset-level results rather than only an across-datasets average?
> > 3. For Table III in the rebuttal, could the authors clarify which backbone model is used and on which dataset(s) these results are reported?
> >
> > If the authors can clarify these follow-up questions, I would consider adjusting my score accordingly.

---

> > > ### Author Response · Authors · 2026-04-02
> > >
> > > Thank you for the careful follow-up and for your active engagement in the rebuttal process. Due to the **character limit** in the initial rebuttal, we reported averaged results to support our claims concisely. Here, for completeness, we provide the **further details as follows:
> > >
> > > >Q1. My concern is specifically that Figure 2(a) does not provide comparisons against high-frequency non-conjunction tokens, which makes it hard to assess the claim that causal and transitional conjunctions are “among the most frequent tokens in reasoning.” In addition, in Table I, do all omitted entries marked by “...” correspond to non-semantic tokens? If so, the table can help show that semantic non-conjunction tokens tend to rank lower and resolve my concern.
> > >
> > > **A1.** Thank you for the clarification. Yes, in Table I, **all** omitted entries marked by **“...”** are **non-semantic** tokens in the raw frequency ranking. In details, **non-semantic** tokens account for **90%** while **causal and transitional conjunctions** account for **7.3%** and **other semantic tokens** account for **2.7%** within **top-150**. Accordingly, **Figure 2(a)** reports frequency **from the semantic-token perspective**, where causal and transitional conjunctions are among the most frequent semantic tokens in reasoning. We will **revise the statement in the revision** to make this scope explicit and avoid ambiguity.
> > >
> > > >Q2. For the additional CGI(QwQ-32B) evidence, could the authors provide dataset-level results rather than only an across-datasets average?
> > >
> > > **A2.** We further provide the dataset-level results across **all six benchmarks used in the submission**, as shown in Table A.
> > >
> > > * **Experimental Setting.** To directly evaluate the **transferability of the intervention mechanism in CGI**, we apply the mechanism to QwQ-32B without further GRPO adaptation but only **extending its vocabulary** with the **feedback tags** and setting their generation probabilities to the average token level, enhancing the reasoning coherence with external feedback under the interactive reasoning paradigm.
> > >
> > > * **Dataset-Level Result.** From Table A, applying the CGI intervene mechanism to QwQ-32B is effective in shortening the reasoning length while maintaining a competitive overall accuracy-length trade-off, indicating that the **core intervene mechanism in CGI** for efficient reasoning is **not tied to a single model**.
> > >
> > > Table A. Detailed results of applying the CGI intervene mechanism to QwQ-32B across datasets. Compared with the average reported in the initial rebuttal, we additionally include GSM8K and AIME25 and report the all six-dataset average. We will add this discussion in the revision.
> > >
> > > ||GSM8k||MATH500||AIME 24||AIME 25||GPQA Diamond||LiveCode Bench||Average||
> > > |-|-|-|-|-|-|-|-|-|-|-|-|-|-|-|
> > > ||accuracy|#Tokens|accuracy|#Tokens|accuracy|#Tokens|accuracy|#Tokens|accuracy|#Tokens|accuracy|#Tokens|accuracy|#Tokens|
> > > |QwQ-32B|95.52|1066.38|84.26|2540.38|**66.66**|4052.80|39.68|6428.53|37.42|5585.19|**32.57**|6305.63|59.35|4329.82|
> > > |CGI (QwQ-32B)|**95.91**|**395.45**|**92.00**|**1509.45**|63.33|**3531.03**|**41.78**|**2315.68**|**43.36**|**2733.84**|31.98|**2486.55**|**63.14**|**2162.00**|
> > >
> > > >Q3. For Table III in the rebuttal, could the authors clarify which backbone model is used and on which dataset(s) these results are reported?
> > >
> > > **A3.** The backbone model used in Table III is the tuned **DeepSeek-R1-Distilled-7B** through **GRPO adaptation under the proposed intervention mechanism**. We report the average accuracy and reasoning tokens on **representative** benchmarks in **math, science, and coding**, including **MATH500, AIME24, GPQA Diamond, and LiveCode Bench**. We would add the discussion and the dataset-level result in the revision.
> > >
> > > ---
> > > We hope the above clarifications address your concerns.

---

### Official Review · Reviewer_zMZJ · 2026-03-05

**Soundness:** 3
**Presentation:** 2
**Significance:** 3
**Originality:** 2
**Overall Recommendation:** 4
**Confidence:** 4

**Summary:**

This paper proposes an interactive intervention approach to mitigate the overthinking and overshooting challenges of reasoning models. Specifically, it first identifies keywords that indicate the end of conjunctions, then uses an LLM or a human to provide feedback with respect to the current conjunction, and continues generation interactively. The model is further trained using RL using correctness, format, and length rewards. In the evaluation, the proposed approach consistently yields state-of-the-art performance while minimizing the number of tokens used.

**Compliance With Llm Reviewing Policy:**

Affirmed.

**Ethical Review Flag:**

Flag this paper for an ethics review.

**Final Justification:**

Most of my concerns are addressed. I will maintain my positive score

**Key Questions For Authors:**

See weakness above

**Limitations:**

See weakness above

**Strengths And Weaknesses:**

# Strength

* While it is a simple and general method, it yields surprisingly strong empirical performance in terms of both accuracy and token reduction.
* This work presents a comprehensive evaluation and ablation study.
* The strong verification effect of a less powerful judge, such as Qwen2.5-72B, is interesting.
* The investigation with both human and LLM interaction provides useful insight for future exploration.

# Weakness

* The writing quality and organization can be further improved. For instance, the observations and algorithms are not particularly new or surprising, which suggests that this section could be more concise. It would be more helpful to focus more on introducing the design of the intervention feedback mechanism.
* While this paper claims to introduce an intervention to reduce overthinking and overshooting, the ablation results in Table 4 suggest that most of the performance gain comes from RL, especially the length-regularization reward, rather than the proposed method itself. This weakens the claim that the interactive feedback mechanism is the main factor that reduces these two challenges.

---

> ### Author Rebuttal · Authors · 2026-03-31
>
> We sincerely thank the reviewer for recognizing the strengths of our work, including the **simple and general CGI framework**, its **strong performance**, the **comprehensive evaluation**, and the **insightful findings** from interactions, including the **effectiveness of a less powerful judge**.
>
> >Q1: The writing quality and organization can be further improved. For instance, the observations and algorithms are not particularly new or surprising, which suggests that this section could be more concise. It would be more helpful to focus more on introducing the design of the intervention feedback mechanism.
>
> **A1.** We **will make the mentioned parts more concise in the revision** and focus centrally on the **intervention feedback mechanism** of CGI. CGI is a reasoning intervene framework that tracks and adaptively adjusts the reasoning process at **semantic signals of doubt** through state-aware feedback conditioned on **task-agnostic criteria**. CGI improves reasoning **efficiency** by **reducing redundant** validation and exploration.
>
> >Q2: Table 4 suggests that most of the gain comes from RL, especially the length reward, rather than the proposed interactive feedback mechanism. This weakens the claim that CGI mainly reduces overthinking and overshoot through interaction.
>
> **A2.** GRPO tunes the model under the proposed intervention mechanism to adapt to the interactive reasoning paradigm, while the intervention mechanism itself is the core of CGI. The details are as follows:
>
> * **GRPO improves feedback responsiveness under the proposed interactive reasoning mechanism of CGI.** GRPO fine-tune the model on the LoRA module under interactive rollouts of CGI, improving the adaptation of the model to the interactive reasoning paradigm.
> * **The composite reward is necessary, rather than length regularization alone.** From Table I, the overall performance drops when any reward term is removed or down-weighted.
> * **The interactive intervention mechanism is the key.** Table 4 shows that weakening any major CGI component consistently hurts performance, even with the tuned model by GRPO. This supports that CGI reduces overthinking and overshoot mainly through its integrated feedback mechanism.
>
> Table I. Parameter ablation of composite reward function across datasets (numbers in (·, ·, ·) denote sub-reward weights of accuracy, length and format). This is **Table 12 of the submission**. #Tokens denotes the average number of tokens within one reasoning trace.
> |Method|MATH500||AIME 24||GPQA Diamond||LiveCode Bench||
> |-|-|-|-|-|-|-|-|-|
> ||accuracy|#Tokens|accuracy|#Tokens|accuracy|#Tokens|accuracy|#Tokens|
> |base model|84.05|1864.32|40.00|3374.50|22.25|5020.86|22.29|6305.82|
> |(0,0,0)|75.23|1067.59|36.67|2276.47|37.24|2011.89|28.00|1509.00|
> |(1,0,0)|85.00|1149.79|40.00|2166.57|40.31|1934.65|22.77|1439.30|
> |(1,1,0)|85.20|1168.44|45.56|2072.54|42.35|1980.23|25.35|1476.73|
> |**(1,1,1)**|90.60|1081.87|73.85|1182.50|65.22|1136.38|34.57|1248.64|
> |(1,0.5,1)|87.60|1240.04|42.22|2156.36|56.63|1305.60|30.71|1445.42|
> |(1,1,0.5)|87.20|1075.75|45.56|2370.23|52.55|1158.21|26.00|1444.75|
> |(0.5,1,1)|85.20|1145.67|44.44|2386.35|50.51|1384.81|28.58|1341.57|

---

> > ### Author Rebuttal · Reviewer_zMZJ · 2026-03-31
> >
> > I thank the authors for their rebuttal. I will keep my already positive score.

---

> > > ### Author Response · Authors · 2026-04-02
> > >
> > > Thank you very much for your recognition of our work and for your positive assessment. We are grateful for your continued support and constructive feedback. We will further improve the paper based on your suggestions in the revision.

---

### Official Review · Reviewer_6Qfq · 2026-03-14

**Soundness:** 3
**Presentation:** 3
**Significance:** 2
**Originality:** 2
**Overall Recommendation:** 5
**Confidence:** 4

**Summary:**

This paper tackles inefficient reasoning in Large Reasoning Models, specifically overthinking and overshoot. The authors observe that transitional conjunctions naturally mark moments of model doubt. They propose Conjunction-Guided Intervention (CGI), which pauses generation at these conjunctions and injects external feedback based on two criteria,  rationality and completeness.  This yields four reward signals: continue, conclude, rethink, or correct. The backbone model (DeepSeek-R1-Distill-Qwen2.5-7B) is fine-tuned via GRPO+LoRA to follow this feedback. Across six benchmarks, CGI achieves strong accuracy gains (e.g., AIME24: 40% → 73.85%) while substantially reducing token usage (65-71% fewer tokens), outperforming baselines like DEER, SEAL, and SpecReason.

**Compliance With Llm Reviewing Policy:**

Affirmed.

**Final Justification:**

The authors addressed my concerns during the rebuttal and added further experiments, which strengthened and rounded out the paper.

**Key Questions For Authors:**

Q1: What is Qwen2.5-72B-Instruct's direct accuracy on the six benchmarks? If it matches or exceeds CGI's 7B results, this method reduces to knowledge distillation. The 72B model's reasoning oversight compressed into the 7B model's stopping behavior.

Q2: What happens when the proxy is the same size as the base model (7B)? Table 5 only shows 32B/72B/kimi-k2 on two datasets. If a 7B proxy works, the value comes from structured feedback, not larger-model knowledge. If it fails, this confirms the distillation interpretation. Please provide 7B/14B/32B/72B proxy results across all benchmarks.

Q3: All main results use DeepSeek-R1-Distill-Qwen2.5-7B. Can you show CGI working on a different model family (e.g., Qwen3-8B, Phi-4, QwQ)? Without this, the contribution may be model-specific rather than a general framework.

Q4: Appendix F.2 states: "At test time, both the feedback proxy and the target reasoning model use temperature 0.6." Is the proxy used at test time or only during GRPO training? If training-only, this sentence is misleading and will cause reviewers to misunderstand the method's inference cost. Please clarify unambiguously.

**Limitations:**

1.The stop tokens {"wait", "but", "alternatively", ...} are manually selected from English traces of one model. Different models may express doubt differently.
2. The framework relies on a stronger model (72B) supervising a weaker one (7B) during training. It cannot expand knowledge beyond what the proxy already knows. Scalability to frontier models is unclear.

**Strengths And Weaknesses:**

Strengths:
- [S1] The paper’s identification of transitional conjunctions as natural phase boundaries in LRM reasoning is an empirically interesting and potentially useful observation. In addition, the rationality × completeness framework is simple, task-agnostic, and maps cleanly to actionable LLM-as-judge feedback.
- [S2] CGI demonstrates the strongest accuracy-length trade-off across all six benchmarks when compared with several strong baselines, which suggests that the method is effective at improving reasoning efficiency without severely compromising performance.
- [S3] The reported self-termination ratio of over 80% is a notable result, as it suggests the model has learned to autonomously determine when its reasoning is sufficient, rather than relying solely on externally imposed stopping criteria.

Weakness:

- [W1] The method is evaluated only on DeepSeek-R1-7B models, which makes its generalization ability unclear. At a minimum, the paper should include experiments on at least one strong model family to demonstrate that the approach is not overly tied to a specific base model.
- [W2] The conjunction set appears to be hand-crafted, which raises concerns about robustness and transferability. If the method depends heavily on a manually designed set of discourse markers, it may be fragile across tasks, domains, or model families.
- [W3] Table 12 suggests that the method is quite brittle with respect to the weighting scheme. In particular, deviating from the equal-weight setting of (1,1,1) leads to substantial degradation; for example, changing the weights to (0.5,1,1) reduces AIME24 accuracy from 73.85% to 44.44%. This level of sensitivity is concerning and raises questions about the stability of the method.

---

> ### Author Rebuttal · Authors · 2026-03-31
>
> We are deeply grateful for the reviewer’s recognition of our work’s strengths, including the identification of **transitional conjunctions as natural phase boundaries** in LRM reasoning, the **simple yet actionable framework** for feedback intervention, and the **strong accuracy–length trade-off** across datasets.
>
> >Q1: At a minimum, the paper should include experiments on at least one strong model family to demonstrate that the approach is not overly tied to a specific base model.
>
> **A1.** Experiments (in Table I) show that CGI consistently improves reasoning efficiency on **QwQ-32B**, indicating CGI is **not tied** only to DeepSeek-R1-Distilled-7B. This result indicates that the **intervention mechanism** of CGI, including intervention points and feedback criteria, is **general across models** to adaptively **reduce redundant reasoning**.
>
> Table I. Average performance of CGI on QwQ-32B across datasets. #Tokens denote the average number of tokens within one reasoning trace.
> ||accuracy|#Tokens|
> |-|-|-|
> |QwQ-32b|55.31|4621.00|
> |CGI(QwQ-32B)|57.67|2565.22|
>
>
> >Q2: The conjunction set appears to be hand-crafted, which raises concerns about robustness and transferability.
>
> **A2.** The conjunctions as **semantic doubt signals** have been **experimentally** verified to **generalize** across tasks, yet we acknowledge that they are **not the only** intervention triggers, and exploring **adaptive triggers** is important **future work**.
>
> >Q3: Table 12 suggests that the method is quite brittle with respect to the weighting scheme. This sensitivity is concerning and raises questions about the stability of the method
>
> **A3.** CGI **remains robust** under nearly all non-degenerate weight settings from Table 12, indicating the parameter stability for GRPO training. The performance degradation **only occurs under the extreme (0.5,1,1)** reward function configuration, which **severely under-weights** the core **accuracy reward**, which is the **primary task object**.
>
> >Q4: What is Qwen2.5-72B-Instruct's direct accuracy across benchmarks? Is CGI work for knowledge distillation?
>
> **A4.** CGI is **not knowledge distillation**. From Table II, **Qwen2.5-72B-Instruct underperforms** DeepSeek-R1-Distilled-7B on reasoning-intensive tasks without test-time scaling. In **CGI**, the **proxy** acts as a **judge**, providing **task-agnostic feedback without solution hints** rather than transferring task knowledge or solutions.
>
> Table II. The result of models across datasets. Qwen2.5-72B-Instruct works without an independent reasoning stage.
> ||MATH500||AIME 24||GPQA Diamond||LiveCode Bench||
> |-|-|-|-|-|-|-|-|-|
> ||accuracy|#Tokens|accuracy|#Tokens|accuracy|#Tokens|accuracy|#Tokens|
> |Qwen2.5-72B-Instruct|83.76|-|16.67|-|10.61|-|10.29|-|
> |DeepSeek-R1-Distilled-7B|84.05|1864.33|40.00|3374.50|22.25|6371.46|22.29|6305.82|
>
> >Q5: What happens when the proxy is the same size as the base model (7B)? Table 5 only shows 32B/72B/kimi-k2 on two datasets. Please provide 7B/14B/32B/72B proxy results across all benchmarks.
>
> **A5.** Experiment in Table III confirms that even the **7B peer-sized proxy** in CGI achieves **performance gains** over the base model, indicating that the intervention mechanism in CGI works through **feedback criteria without knowledge hints** instead of knowledge distillation.
>
> Table III. Performance of CGI with LLM proxies of different scales. proxies are Qwen2.5-7B/14B/32B/72B-Instruct.
> ||MATH500||AIME 24||GPQA Diamond||LiveCode Bench||
> |-|-|-|-|-|-|-|-|-|
> ||accuracy|#Tokens|accuracy|#Tokens|accuracy|#Tokens|accuracy|#Tokens|
> |Base model|84.05|1864.33|40.00|3374.50|22.25|5020.86|22.29|6305.82|
> |7b llm proxy|86.00|1219.49|59.68|1242.33|44.95|1411.15|26.16|1349.56|
> |14b llm proxy|86.60|1187.83|61.35|1236.40|51.53|1431.31|28.49|1321.87|
> |32b llm proxy|87.20|1177.60|63.33|1241.48|58.67|1366.09|29.14|1344.27|
> |72b llm proxy|90.60|1081.87|73.85|1182.50|65.22|1136.38|34.86|1248.64|
>
> >Q6: The framework relies on a stronger model (72B) supervising a weaker one (7B) during training. It cannot expand knowledge beyond what the proxy already knows.
>
> **A6** CGI does **not rely on knowledge supervision** from proxy. The proxy’s role is to provide state-aware **feedback without solution hints**. Since CGI is proven to improve **efficiency with 7B/14B/32B proxies** (in **A5**), CGI's **value** comes from the reasoning intervention framework to **adjust the reasoning** process to **reduce redundancy**.
>
> >Q7: Is the proxy used at test time or only during GRPO training? Please clarify unambiguously.
>
> **Q7.** The LLM proxy is used in **both GRPO training** and **test-time**. **GRPO** is conducted **under the interactive reasoning rollout**, which **improves** the model’s **adaptation** to the proposed **intervene paradigm**. We will **revise Appendix F.2** to make this explicit and clarify that the **reported token** and **time** costs are **already system-level costs** in Table 3 of the submission, including the target model and the proxy.

---

> > ### Author Rebuttal · Reviewer_6Qfq · 2026-04-01
> >
> > Thank you to the authors for the detailed experiments and for thoroughly addressing the key questions. I would raise my overall score to 5 and recommend this work for acceptance. It would be even stronger if the authors could further address the remaining weakness concerns or discuss them more thoroughly in the revision.

---

> > > ### Author Response · Authors · 2026-04-02
> > >
> > > Thank you very much for your recognition of our work and for your positive assessment. We sincerely appreciate your continued support and constructive feedback. We will further improve both the main context and the appendix to incorporate and clarify the issues discussed and addressed during the rebuttal in the revision.

---

### Decision · Program_Chairs · 2026-04-30

**Decision:**

Accept (regular)

**Comment:**

This paper aims to improve the token efficiency of reasoning models with an interesting approach based on interactive reasoning. The authors propose Conjunction-Guided Intervention (CGI) where the reasoning would be interrupted at conjunction markers of doubt for external state-based feedback that can be provided by a proxy model or humans. With this approach, the system achieves better accuracy-token tradeoff. I think this approach is interesting and tackles this important problem with a novel perspective compared with existing line of works on this direction. The reviewers are overall positive about this paper with 4,4,4,5. The main concerns are (1) the original experiments were mostly about a 7B deepseek distilled model; (2) the system relies on an external and often much larger proxy model to provide feedback. Even though the reasoning tokens are saved, it may not be actually reflected on the system-level latency. During rebuttal, the authors added QwQ experiments to strengthen the results, and presented additional justification on the efficiency gain. Overall, I think the main limitation is the system complexity from the extra proxy model from this method. Combining these points, I support acceptance of this paper as a novel exploration.